# Altered glycolysis triggers impaired mitochondrial metabolism and mTORC1 activation in diabetic β-cells

Elizabeth Haythorne [1] ✉, Matthew Lloyd [1], John Walsby-Tickle [2], Andrei I. Tarasov[3], Jonas Sandbrink[1], Idoia Portillo[1], Raul Terron Exposito[1,5], Gregor Sachse [1,6], Malgorzata Cyranka[1], Maria Rohm [1,5], Patrik Rorsman [4], James McCullagh [2] & Frances M. Ashcroft [1] ✉

Chronic hyperglycaemia causes a dramatic decrease in mitochondrial metabolism and insulin content in pancreatic β-cells. This underlies the progressive decline in β-cell function in diabetes. However, the molecular mechanisms by which hyperglycaemia produces these effects remain unresolved. Using isolated islets and INS-1 cells, we show here that one or more glycolytic metabolites downstream of phosphofructokinase and upstream of GAPDH mediates the effects of chronic hyperglycemia. This metabolite stimulates marked upregulation of mTORC1 and concomitant downregulation of AMPK. Increased mTORC1 activity causes inhibition of pyruvate dehydrogenase which reduces pyruvate entry into the tricarboxylic acid cycle and partially accounts for the hyperglycaemia-induced reduction in oxidative phosphorylation and insulin secretion. In addition, hyperglycaemia (or diabetes) dramatically inhibits GAPDH activity, thereby impairing glucose metabolism. Our data also reveal that restricting glucose metabolism during hyperglycaemia prevents these changes and thus may be of therapeutic benefit. In summary, we have identified a pathway by which chronic hyperglycaemia reduces β-cell function.

Type 2 diabetes (T2D) is a serious global health problem. It is characterised by defective insulin secretion from the β-cells of the pancreatic islets, which results in chronically elevated blood glucose. T2D normally presents in later adult life and is a progressive disease that begins with impaired glucose tolerance and advances to diabetes as β-cells gradually fail. By the time of diagnosis as much as 50% of β-cell function has been lost[1]. There is accumulating evidence that β-cell decline is primarily driven by increasing hyperglycaemia, which results in both a dramatic loss of insulin content and a decrease in β-cell

metabolism[2–5]. Although there is some loss of β-cells, this is too small to account for the reduction in insulin release[6,7], or the reversal of T2D following bariatric surgery or a low-calorie diet[8,9]. Furthermore, as insulin secretion in T2D can be enhanced by drugs that bypass the metabolic steps (such as sulphonylureas or GLP-1-based therapies), it appears insulin content is also not limiting and thus that metabolic failure plays a critical role in diabetes development.

Glucose metabolism is essential for insulin secretion, coupling changes in blood glucose to fluctuations in insulin release. In non-

[1]Department of Physiology, Anatomy and Genetics and OXION, University of Oxford, Parks Road, Oxford OX1 3PT, UK. [2]Chemistry Research Laboratory, Department of Chemistry, University of Oxford, Mansfield Road, Oxford OX1 3TA, UK. [3]School of Biomedical Sciences, Ulster University, Coleraine BT52 1SA Northern Ireland, UK. [4]Oxford Centre for Diabetes, Endocrinology and Metabolism, University of Oxford, Churchill Hospital, Oxford OX3 7LJ, UK. [5]Present address: Institute for Diabetes and Cancer (IDC), Helmholtz Center, Munich, Neuherberg 85764, Germany. [6]Present address: Brandenburg Medical School (Theodor Fontane), ZTM-BB, Brandenburg a. d. H 14770, Germany. ✉e-mail: Elizabeth.Haythorne@dpag.ox.ac.uk; frances.ashcroft@dpag.ox.ac.uk

diabetic β-cells, glucose uptake and its subsequent phosphorylation by glucokinase drive glycolysis and pyruvate generation[10]. Pyruvate enters mitochondria, where it is oxidised in the tricarboxylic acid (TCA) cycle, generating reducing equivalents (NADH and $FADH_2$) that are utilised by the electron transport chain (ETC) for ATP synthesis. Glucose-stimulated insulin secretion (GSIS) is impaired by inhibitors of mitochondrial metabolism, mitochondrial uncouplers or mitochondrial dysfunction, emphasising the pivotal role of mitochondrial metabolism[11]. Elevation of the intracellular ATP/ADP ratio closes ATP-sensitive $K^+$ ($K_{ATP}$) channels in the β-cell plasma membrane, triggering depolarisation and $Ca^{2+}$-dependent electrical activity[12]. The resulting rise in cytoplasmic $Ca^{2+}$ initiates exocytosis of insulin granules. Marked changes in mitochondrial metabolism and in metabolic gene and protein expression have been identified in islets isolated from mouse models of diabetes[2,5,13,14], in human islets from patients with T2D[15–17] and in β-cell lines exposed to chronic hyperglycaemia[5,18].

How hyperglycaemia impairs β-cell metabolism is unclear, but one possibility is via activation of the mechanistic target of rapamycin complex 1 pathway (mTORC1), a master regulator of metabolism, growth and survival[19]. When activated by nutrients, growth factors or intracellular signals, mTORC1 switches metabolism from a catabolic to an anabolic state by stimulating protein, lipid, nucleotide and ATP synthesis. mTORC1 promotes protein synthesis by phosphorylating two key downstream substrates, the ribosomal protein S6 kinase (S6K) and the eIF4E binding protein, 4E-BP1[19]. It also governs de novo lipid biosynthesis and glycolysis by regulating transcription factors such as SREBP1/2 and HIF-1α, respectively[20].

Although mTORC1 is essential for β-cell survival and proliferation under physiological conditions, there is evidence that its hyperactivation contributes to impaired β-cell function in diabetes[21]. Thus, mTORC1 activity is markedly increased in human islets, mouse islets and β-cell lines cultured at high glucose, as well as in islets isolated from various mouse models of T2D and from patients with T2D[22–25]; furthermore, short-term inhibition of the mTORC1-S6K signalling pathway restored insulin secretion in human diabetic β-cells[23]. In addition, in mouse embryonic fibroblasts, mTORC1 promoted a shift in glucose metabolism from oxidative phosphorylation to glycolysis, and led to increased flux through the pentose phosphate pathway, by stimulating the expression of genes encoding nearly every step of glycolysis and the pentose phosphate pathway[20]. These gene changes resemble those seen in diabetic islets and INS-1 cells exposed to chronic hyperglycaemia[5]. Finally, mTORC1 is an important inhibitor of autophagy[21], which is downregulated in islets from diabetic mice[13] and human patients[26], or β-cells cultured at high glucose[27]. It therefore seems possible that diabetes mediates its deleterious effects on β-cell function, at least in part, by mTORC1 activation. However, this remains to be tested. How hyperglycaemia might cause mTORC1 activation is also unknown.

Here, we investigated the mechanism(s) by which chronic hyperglycaemia and diabetes lead to impaired β-cell metabolism, and explored if these are linked to mTORC1 activation. We provide evidence that a glycolytic metabolite downstream of phosphofructokinase and upstream of glyceraldehyde-3-phosphate dehydrogenase (GAPDH) is instrumental in mediating the effects of diabetes and chronic hyperglycaemia on β-cell metabolism. This occurs, in part, through marked upregulation of mTORC1, which leads to changes in metabolic gene expression, oxidative phosphorylation and insulin secretion. In addition, we show diabetes dramatically inhibits the activity of GAPDH and pyruvate dehydrogenase (PDH), impairing both glycolytic metabolism and pyruvate entry into the TCA cycle. Our results support the idea that progressive impairment of β-cell metabolism, induced by increasing hyperglycaemia, speeds T2D development. In addition, we provide evidence that suggests reducing glycolysis, at the level of glucokinase, during chronic hyperglycaemia, may slow diabetes progression.

## Results

To explore the effects of chronic hyperglycaemia on β-cell metabolism, we used two models. First, the insulin-secreting cell line INS-1 832/13 cells (INS-1 cells), cultured either at 5 mM glucose (low glucose, LG-cells) or 25 mM glucose (high glucose, HG-cells) for 48 h. Second, islets isolated from diabetic βV59M mice, a model of human neonatal diabetes (ND). These mice selectively express an inducible activating ND $K_{ATP}$ channel mutation (Kir6.2-V59M) in their β-cells that rapidly switches off insulin secretion following tamoxifen injection[2]. They exhibit hyperglycaemia and hypoinsulinaemia but not dyslipidaemia and thus provide an in vivo model of chronic hyperglycaemia in the absence of obesity or dyslipidaemia. We refer to them as diabetic mice (and their islets as diabetic islets). The β-cell changes found in diabetic βV59M mice are prevented by restoration of euglycaemia with insulin, indicating they are due to hyperglycaemia/hypoinsulinaemia not $K_{ATP}$ channel activation per se[2].

### The effects of chronic hyperglycaemia require glucose metabolism

To determine if glucose itself or one of its downstream metabolites mediates the deleterious effects of chronic hyperglycaemia, we partially inhibited glucokinase with mannoheptulose[28,29] (Fig. 1; Supplementary Fig. 1a, b). Glucokinase (GCK) catalyses glucose phosphorylation, the first step in glucose metabolism (Fig. 1a). We cultured LG-cells and HG-cells, and control and diabetic islets, with or without 10 mM mannoheptulose for 48 h. We subsequently removed the drug for the measurement of insulin secretion.

Co-culture with mannoheptulose had little effect on glucose-stimulated insulin secretion (GSIS) or insulin content in LG-cells, but largely prevented the dramatic reduction in GSIS and insulin content produced by chronic hyperglycaemia (Fig. 1b, c). It also prevented the decrease in the glucose-stimulated oxygen-consumption rate, the reduction in ATP-linked respiration and the increase in mitochondrial leak found in HG-cells (Fig. 1d, e). Most of the changes in metabolic gene expression produced by chronic hyperglycaemia were also prevented (Fig. 1f, g).

In non-diabetic (control) islets, co-culture with mannoheptulose had no effect on GSIS (measured after removal of the inhibitor) but markedly reduced insulin content (Supplementary Fig. 1c, d). As previously reported[2,30], diabetic islets had a lower insulin content than control islets and largely failed to respond to 20 mM glucose with insulin secretion. Insulin content was not restored by mannoheptulose, suggesting that the drug can prevent the fall in insulin content (in INS-1 cells) but may not reverse it (in isolated islets) on the time scale of our experiments. Insulin secretion, however, partially recovered, consistent with the partial restoration of metabolism in diabetic β-cells. Full recovery of insulin secretion is not expected, due to the hyperpolarising $K_{ATP}$ channel mutation, but some recovery is anticipated because the Kir6.2-V59M mutation does not fully prevent $K_{ATP}$ channel closure in response to glucose[30].

Taken together, these data suggest that glucose-6-phosphate (G6P) or a downstream metabolite, rather than glucose itself, mediates most of the effects of chronic hyperglycaemia on insulin content, β-cell metabolism and insulin secretion. They also demonstrate that partial inhibition of glucose metabolism protects INS-1 cells from the deleterious effects of chronic hyperglycaemia, and can partially restore GSIS in diabetic islets.

### A glycolytic metabolite mediates the deleterious effects of chronic hyperglycaemia

To distinguish if a glycolytic or a mitochondrial metabolite mediates the effects of chronic hyperglycaemia, we examined the effects of substrates metabolised entirely within the mitochondria (Fig. 2a). We used the methyl ester form of pyruvate (Me-pyruvate) as β-cells express negligible levels of the monocarboxylate transporter,

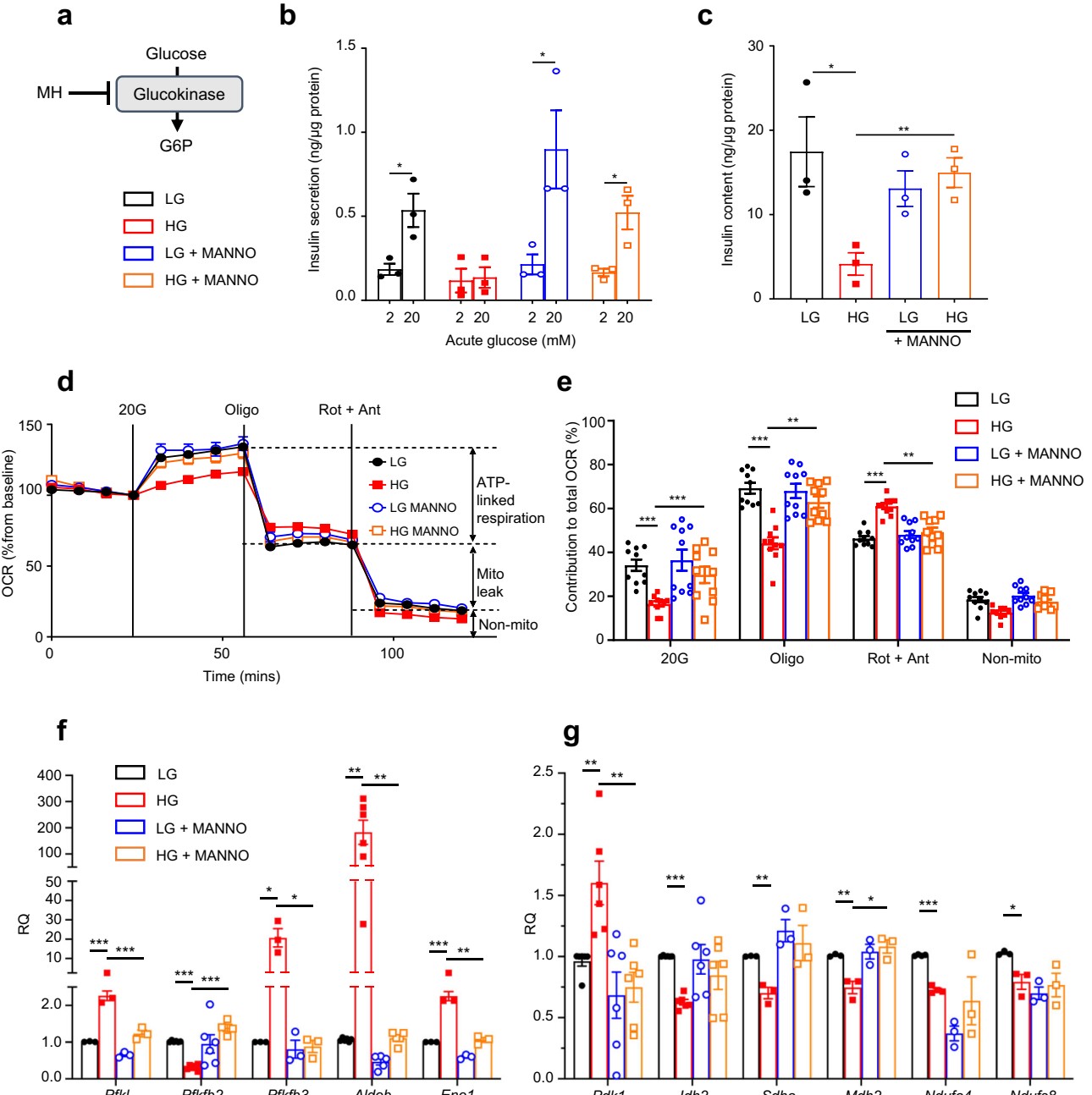

**Fig. 1 | Inhibition of glucokinase prevents the effects of chronic hyperglycaemia. a** Schematic showing how mannoheptulose (MH) inhibits glucose metabolism. **b, c** Insulin secretion (**b**) and insulin content (**c**) in LG-cells and HG-cells cultured for 48 h ± 10 mM mannoheptulose (MANNO) and then stimulated with 2 mM or 20 mM glucose. Mannoheptulose was omitted during the assay (*n* = 3 biologically independent experiments). **d** Oxygen-consumption rate (OCR) in LG-cells and HG-cells cultured for 48 h ± 10 mM MANNO. OCR was recorded at 2 mM glucose and after sequential addition of 20 mM glucose (20 G), 1 μM oligomycin (Oligo) and 0.5 μM rotenone + 0.5 μM antimycin A (Rot + Ant). Data are expressed as the percentage change from baseline (2 mM glucose); *n* = 10 biologically independent experiments per group. **e** Percentage change in OCR when glucose was raised from 2 to 20 mM (20 G), ATP-linked OCR (Oligo), OCR required to maintain the mitochondrial leak (Rot + Ant) and non-mitochondrial OCR (non-mito); *n* = 10

biologically independent experiments per group. Same data as in (**d**). **f, g** mRNA levels of the indicated genes involved in glycolytic (**f**) and mitochondrial (**g**) metabolism as assessed by qPCR in LG-cells and HG-cells cultured for 48 h ± 10 mM mannoheptulose (*Pdk1, Idh2* and *Ndufs8*, *n* = 6 biologically independent experiments; *Ndufa4*, *n* = 4 biologically independent experiments; *Pfkl, Pfkfb3, Eno1, Sdha* and *Mdh2*, *n* = 3 biologically independent experiments; *Aldob*, *n* = 6 biologically independent experiments for LG and HG but *n* = 5 for LG + MANNO and HG + MANNO; *Ndufs8*, *n* = 6 biologically independent experiments for LG but *n* = 3 for LG + MANNO, HG and HG + MANNO). All panels show individual data points and mean ± s.e.m. \**P* < 0.05, \*\**P* < 0.01, \*\*\**P* < 0.001; two-tailed unpaired Student's *t* test. LG-cells (black), HG-cells (red), LG-cells + mannoheptulose (blue), HG-cells + mannoheptulose (orange). Source data are provided as a Source Data file.

MCT1[31,32], and Me-pyruvate, but not pyruvate, stimulates insulin release[33]. Unlike chronic hyperglycaemia, chronic Me-pyruvate exposure (48 h, in 0 mM glucose) did not attenuate GSIS or reduce insulin content in INS-1 cells (Fig. 2b, c), as reported previously[18]. Chronic Me-pyruvate also did not mimic the effects of high glucose on metabolic

gene expression; most genes examined were unaffected or showed changes in the opposite direction (e.g., *Aldob, Idh2*) (Fig. 2d, e). These data argue that the key metabolite responsible for the effects of chronic hyperglycaemia on β-cell metabolism and insulin content must lie between G6P and pyruvate. We excluded a role for the

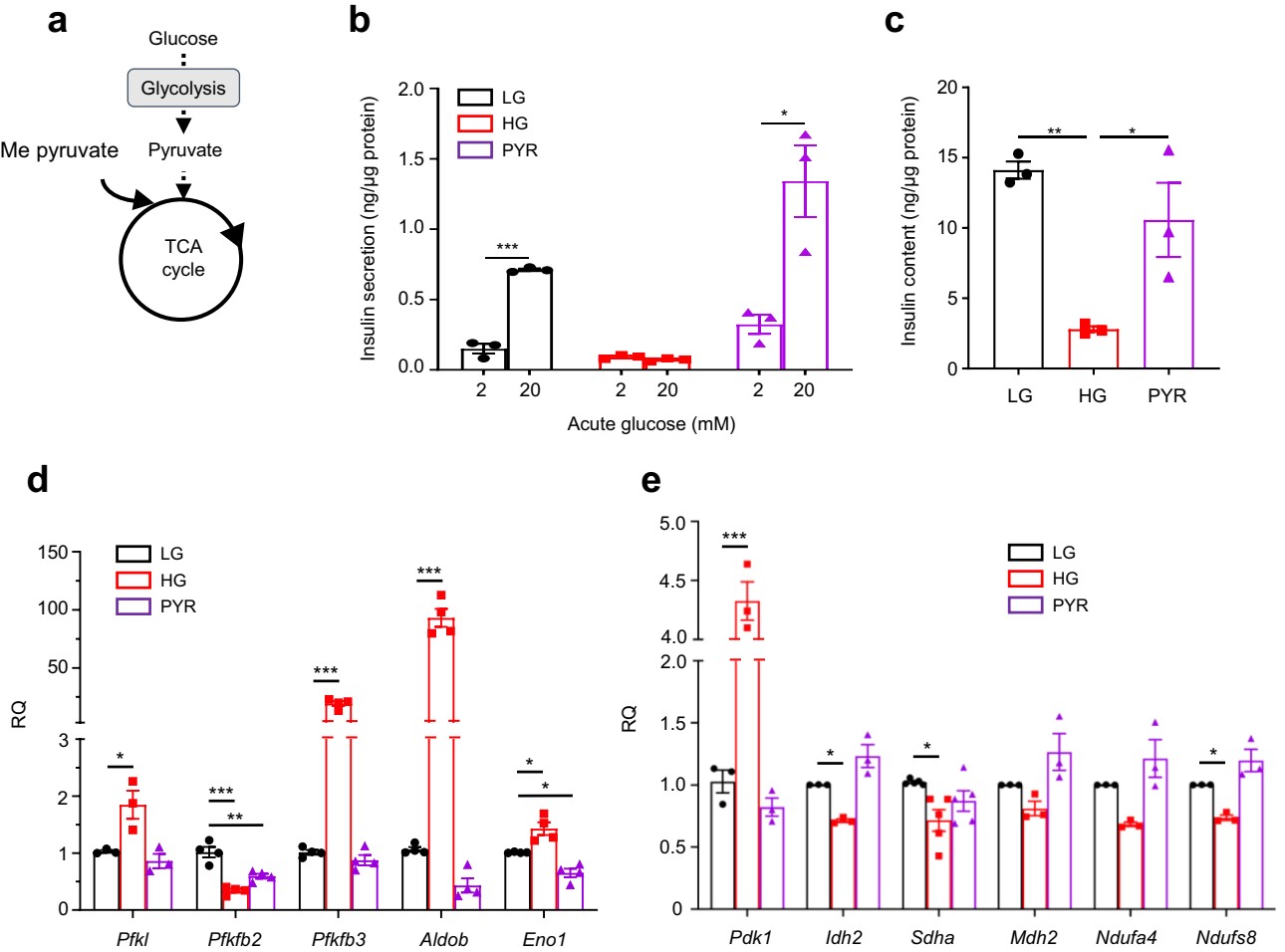

**Fig. 2 | Chronic effects of the mitochondrial substrate pyruvate on insulin secretion. a** Schematic showing where methyl pyruvate (Me-pyruvate) enters metabolism. **b, c** Insulin secretion (**b**) and insulin content (**c**) in LG- and HG-cells, or cells cultured with 20 mM methyl pyruvate for 48 h (PYR) (*n* = 3 biologically independent experiments). **d, e** mRNA levels for the indicated genes involved in glycolytic (**d**) and mitochondrial (**e**) metabolism assessed by qPCR in LG- and HG-cells, or cells cultured with 20 mM methyl pyruvate for 48 h (*Pfkl, Pdk1, Idh2, Mdh2,*

*Ndufa4* and *Ndufs8*, *n* = 3 biologically independent experiments; *Pfkfb2, Pfkfb3, Aldob* and *Eno1*, *n* = 4 biologically independent experiments; *Sdha*, *n* = 5 biologically independent experiments). All panels show individual data points and mean ± s.e.m. *$P < 0.05$, **$P < 0.01$, ***$P < 0.001$, two-tailed unpaired Student's *t* test. LG-cells (black), HG-cells (red), PYR-cells (purple). Source data are provided as a Source Data file.

pentose phosphate pathway as inhibition of 6-phosphogluconate dehydrogenase with 6-aminonicotinamide (6-AN) did not prevent the effects of hyperglycaemia on insulin secretion, insulin content or gene expression (Supplementary Fig. 2).

**Metabolomics analysis of INS-1 cells and islets**

We performed discovery metabolomics analysis using both INS-1 cells and islet models to determine which glycolytic metabolites were altered by chronic hyperglycaemia. The abundance of numerous glycolytic and TCA cycle intermediates differed significantly between LG-cells and HG-cells, and between control and diabetic islets, both at basal glucose (2 mM) and when stimulated with 20 mM glucose for 1 h (Fig. 3 and Supplementary Data). In diabetic islets, the most striking changes at 2 mM glucose were in the relative abundances of fructose, fructose-1,6-bisphosphate (F1,6BP), fructose-2,6-bisphosphate (F2,6BP) and pyruvate: all were substantially increased in comparison to control islets (Fig. 3a, b and Supplementary Data 1B). Dihydroxyacetone phosphate (DHAP) was also increased in diabetic islets when measured biochemically (Fig. 3a). At 20 mM glucose, large increases in fructose-6-phosphate (F6P), F1,6BP, F2,6BP and some pentose phosphate pathway intermediates were observed (Fig. 3a and Supplementary Data 1C). Most TCA cycle metabolites were

downregulated in diabetic islets at 2 mM and 20 mM glucose (Fig. 3a). Similar changes were seen in HG-cells (Supplementary Data 2A, B). Thus, one or more of the altered metabolites may mediate the effects of chronic hyperglycaemia on glucose metabolism.

As the observed increase in metabolite abundance could reflect either increased production or reduced consumption, we measured the activity of the glycolytic enzymes involved. Phosphofructokinase (PFK) (Fig. 3c), fructose bisphosphatase (FBPase) (Fig. 3d) and aldolase activity (Fig. 3e) were all elevated in diabetic islets when compared to control islets. Remarkably, however, the activity of glyceraldehyde-3-phosphate dehydrogenase (GAPDH) was dramatically downregulated in diabetic islets (Fig. 3f), despite a threefold increase in GAPDH protein[5]. We also observed a significant reduction in GAPDH activity in HG-cells compared to LG-cells (Supplementary Fig. 3a). The changes in enzyme activity identified were commensurate with the observed increase in abundance of F6P, F1,6BP and DHAP.

We next investigated whether GAPDH inhibition alone was sufficient to cause changes in glucose metabolites and insulin secretion resembling those produced by chronic hyperglycaemia. Acute application of the GAPDH inhibitor koningic acid (KA, 5 µM) caused a significant reduction in GAPDH activity but only slightly reduced insulin secretion in LG-cells (Supplementary Fig. 3b, c). However, culturing LG-

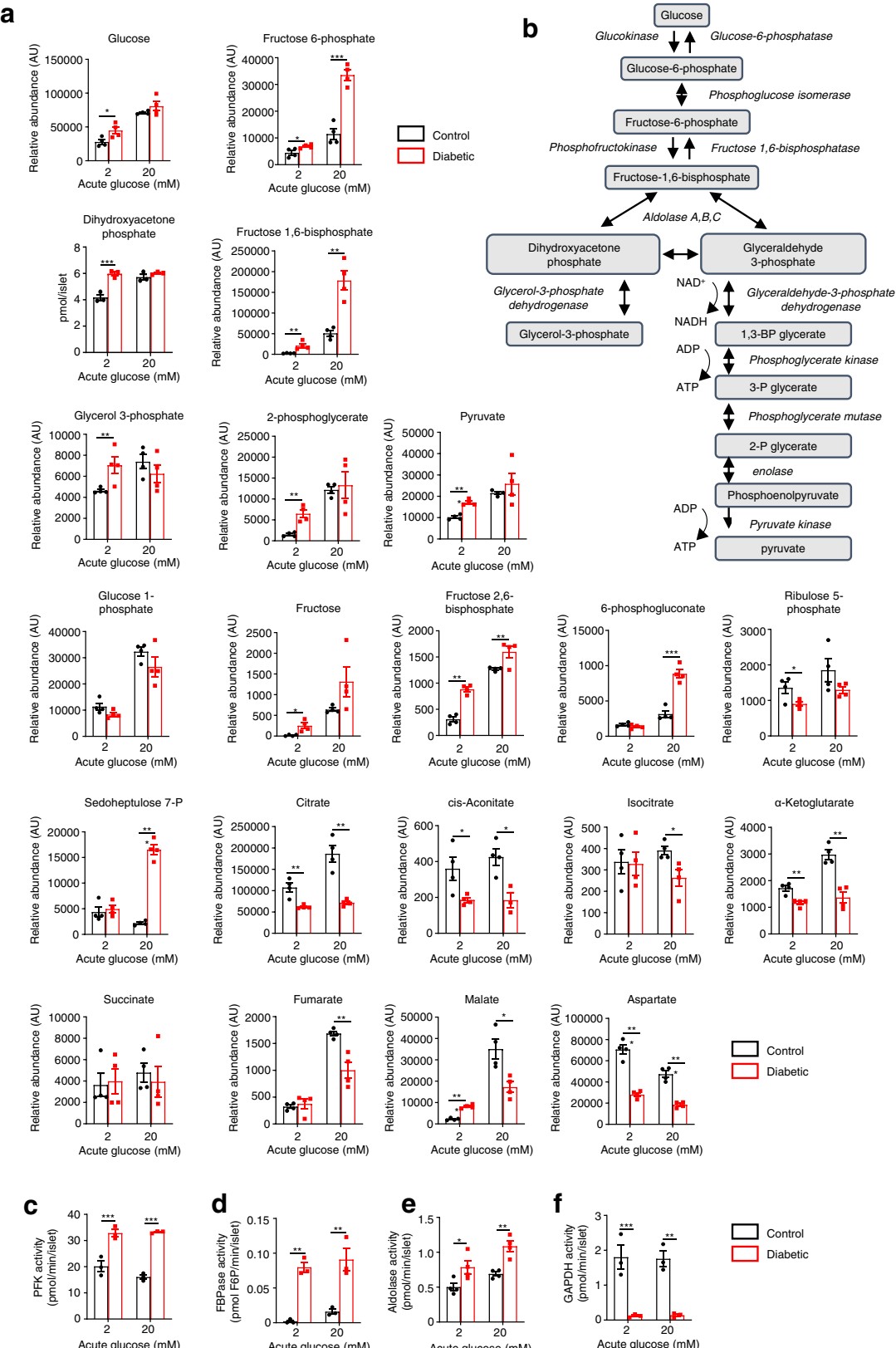

**Fig. 3 | Changes in metabolite abundance induced by chronic hyperglycaemia.**
**a** Abundances of selected glycolytic, pentose phosphate pathway and TCA cycle metabolites in control (black bars) and diabetic (red bars) islets stimulated with 2 mM or 20 mM glucose. ($n$ = 4 animals/genotype). Metabolites were measured by mass spectrometry, except for DHAP which was measured with a fluorimetric assay. **b** Schematic of glycolysis. **c**–**f** Activity of the indicated glycolytic enzymes in control (black) and diabetic (red) islets stimulated with 2 or 20 mM glucose.

**c** Phosphofructokinase (PFK, $n$ = 3 animals/genotype). **d** Fructose-1,6-bisphosphatase (FBPase, $n$ = 3 animals/genotype). **e** Aldolase ($n$ = 4 animals/genotype). **f** Glyceraldehyde-3-phosphate dehydrogenase (GAPDH, $n$ = 3 animals/genotype). All panels show individual data points and mean ± s.e.m. *$P$ < 0.05, **$P$ < 0.01, ***$P$ < 0.001, two-tailed unpaired Student's $t$ test. Source data are provided as a Source Data file, and relative abundances of all metabolites identified are provided as Supplementary Data file 1.

cells with KA for 48 h, produced a striking increase in the abundance of metabolites upstream of GAPDH and a decrease in downstream metabolites such as TCA cycle intermediates, when measured at either 2 mM or 20 mM glucose (Fig. 4a, b and Supplementary Fig. 3d). At 2 mM (basal) glucose, F6P, F1,6BP and NADH were significantly increased, and the increase in the combined abundance of DHAP and glyceraldehyde-3-phosphate (GA3P) was close to significance. At 20 mM glucose, F6P, F1,6BP and DHAP/GA3P were substantially elevated in abundance, and the abundance of many TCA cycle metabolites was reduced (e.g., pyruvate, citrate, malate, NADH) (Fig. 4a, b). These results are consistent with the hypothesis that GAPDH inhibition contributes to the metabolic changes induced by chronic hyperglycaemia. GSIS and insulin content were reduced following culture of LG-cells with KA (Fig. 4c, d). Changes in mRNA expression resembling those caused by chronic hyperglycaemia were also observed, although their magnitude was generally smaller (Fig. 4e, f).

### Chronic hyperglycaemia inhibits AMPK and activates mTORC1

Taken together, our results suggest that one or more metabolites upstream of GAPDH mediate the effects of chronic hyperglycaemia on glucose metabolism. How, then, might chronic elevation of such a metabolite lead to impaired β-cell metabolism? One possibility is via activation of mTORC1, which is hyperactivated in islets from patients with T2D[23] and which, in other cell types, has been shown to be activated by F1,6BP[34], DHAP[35] and GA3P[36].

We therefore explored the effect of chronic hyperglycaemia on the activity of mTORC1 and its upstream regulator AMPK. AMPK activity was determined by phosphorylation of AMPKα at threonine 172 and of Raptor at serine 792, and mTORC1 activity by phosphorylation of its downstream targets, the ribosomal protein S6 (at serine 240/244) and 4E-BP1 (at threonine 37/46). AMPK was active at 2 mM glucose in both control islets (Fig. 5a–c) and LG-cells (Fig. 5f–h), but was inhibited by acute glucose elevation to 20 mM. In contrast, AMPK was largely inactive in diabetic βV59M islets and in HG-cells at both 2 and 20 mM glucose. Conversely, mTORC1 was stimulated by 20 mM glucose in control islets but its activity was already elevated at 2 mM glucose in diabetic islets and was not further increased by 20 mM glucose (Fig. 5a, d, e). HG-cells also displayed hyperactivation of mTORC1 signalling at 2 mM glucose compared to LG-cells (Fig. 5f, i, j). Thus, chronic hyperglycaemia in both INS-1 cells and islets was associated with concurrent activation of mTORC1 and inhibition of AMPK. Consistent with the idea that the key metabolite lies upstream of GAPDH, 48 h culture with KA, or knockdown of GAPDH, in LG-cells mimicked the effects of chronic hyperglycaemia on AMPK and mTORC1 activity (Fig. 5k–m and Supplementary Fig. 4a, d–f).

To ascertain if AMPK activity in β-cells is predominantly governed by changes in cellular adenine nucleotide levels (ATP, ADP, AMP)[37] or, as suggested in other cell types, by a glycolytic metabolite[34,38,39], we compared the acute response to glucose and mitochondrial metabolites. Both the ATP/ADP ratio and insulin secretion were elevated by acute stimulation with 20 mM glucose, Me-pyruvate, leucine or Me-succinate (Fig. 6a, b). However, glucose was the only substrate that inhibited AMPK phosphorylation (Fig. 6c, d). It can therefore be argued that AMPK activity in β-cells is regulated by glucose via a pathway that is independent of intracellular nucleotide levels, and that both the acute and chronic AMPK response to glucose involve a glycolytic metabolite. Both Me-pyruvate and leucine stimulated mTORC1 when applied acutely (Fig. 6e). In contrast, chronic incubation with 20 mM Me-pyruvate or leucine did not result in sustained activation of mTORC1 (Supplementary Fig. 5a–f). Thus, sustained activation of mTORC1 is specific to glucose.

Our data support the idea that a glycolytic metabolite lying between glucokinase and GAPDH mediates the effect of chronic hyperglycaemia on both mTORC1 and AMPK. We reasoned that knockdown of an enzyme upstream of the key metabolite would prevent mTORC1 activation and AMPK inhibition in HG-cells, whereas knockdown of a downstream enzyme might cause their reciprocal activation/inhibition in LG-cells. As F1,6BP[34], DHAP[35] and GA3P[36], as well as PFK itself[40], have all been implicated in regulating mTORC1 activity, we next knocked down PFK in LG- and HG-cells (Supplementary Fig. 4b). The combined knockdown of *Pfkl* and *Pfkm* prevented the phosphorylation of S6 and the reduction in AMPK phosphorylation in HG-cells (Fig. 6f–h). We also tested the effect of chronic PFK-15, an inhibitor of PFKFB3 (6-phosphofructose-2-kinase or PFK2) and thereby an indirect inhibitor of PFK[41]. Chronic PFK-15 recapitulated the effects of PFK knockdown in HG-cells (Fig. 6i–k). These data indicate that the metabolite that mediates the reciprocal activation of mTORC1 and inhibition of AMPK in chronic hyperglycaemia lies between PFK and GAPDH (i.e., F1,6BP, GA3P or DHAP).

Expression of aldolase B is strikingly upregulated in HG-cells (Fig. 1f), in βV59M diabetic islets[5], and in islets from T2D donors[17]. Although we were unable to completely knock down aldolase B, we found that partial knockdown of both aldolase A + B led to partial upregulation of AMPK activity (Supplementary Fig. 4c, g, h). This is consistent with a role for aldolase/F1,6BP in AMPK regulation. However, we did not see a reciprocal inhibition of S6 kinase (Supplementary Fig. 4g, i). This suggests a different mechanism may be involved in mTORC1 regulation.

### Effect of inhibiting S6K activity on β-cell function

mTORC1 has multiple downstream effects, many of which are mediated by S6 kinase (S6K)[19]. To determine if S6K is involved in mediating the effects of chronic hyperglycaemia, we inhibited S6K with 10 μM PF-4708671[42], which reduced ribosomal S6 phosphorylation by ~50% and restored AMPK phosphorylation in HG-cells to control levels (Fig. 7a–c). PF-4708671 largely prevented the detrimental effects of chronic hyperglycaemia on GSIS (Fig. 7d) and, partially, on insulin content (Fig. 7e). A second inhibitor of S6K (LY2584702[43]) produced a similar effect on glucose-stimulated insulin secretion in HG-cells (Supplementary Fig. 6a, b).

Chronic PF-4708671 restored the reduction in glucose-stimulated oxygen-consumption rate (OCR) of INS-1 cells produced by chronic hyperglycaemia (Fig. 8a, b) and partially restored (<50%) ATP-linked respiration. However, the increased mitochondrial leak (as demonstrated by the OCR sensitive to inhibition by rotenone−antimycin) and the higher basal respiration observed in HG-cells were not normalised (Fig. 8b, e and Supplementary Fig. 6c). Some hyperglycaemia-induced changes in metabolic gene expression were reduced, particularly those of glycolytic genes (Supplementary Fig. 6e). However, with the possible exception of *Pdk1*, all mitochondrial genes examined were unaffected by S6K inhibition (Supplementary Fig. 6f).

Culture of 2-week diabetic islets with PF-4708671 for 48 h normalised both AMPK and mTORC1 signalling (Fig. 7f–h). It did not restore the reduced insulin content but partially increased GSIS in diabetic islets (Fig. 7i, j); full restoration is not expected because of the activating $K_{ATP}$ channel mutation (see above) and the reduced insulin content. It is likely that some of the differences between diabetic islets and HG-cells are due to the much longer period of chronic hyperglycaemia (2 weeks vs. 2 days), or because it is easier to prevent changes (INS-1 cells) than to reverse them (diabetic islets).

Chronic hyperglycaemia caused a dramatic fall in both basal and glucose-stimulated OCR in islets (Fig. 8c, d, f), as previously reported[5]. Culture of diabetic islets with PF-4708671 for 48 h had no effect on basal OCR (Fig. 8f), but produced a striking increase in glucose-stimulated OCR and in ATP-linked respiration (Fig. 8d). Nevertheless, absolute levels remained far below those of control islets (Supplementary Fig. 6d). Thus, S6K inhibition did not fully reverse the effects of chronic hyperglycaemia on mitochondrial metabolism, either because S6K was not fully inhibited or because other, as yet unidentified, mechanisms are involved. Taken together, these data

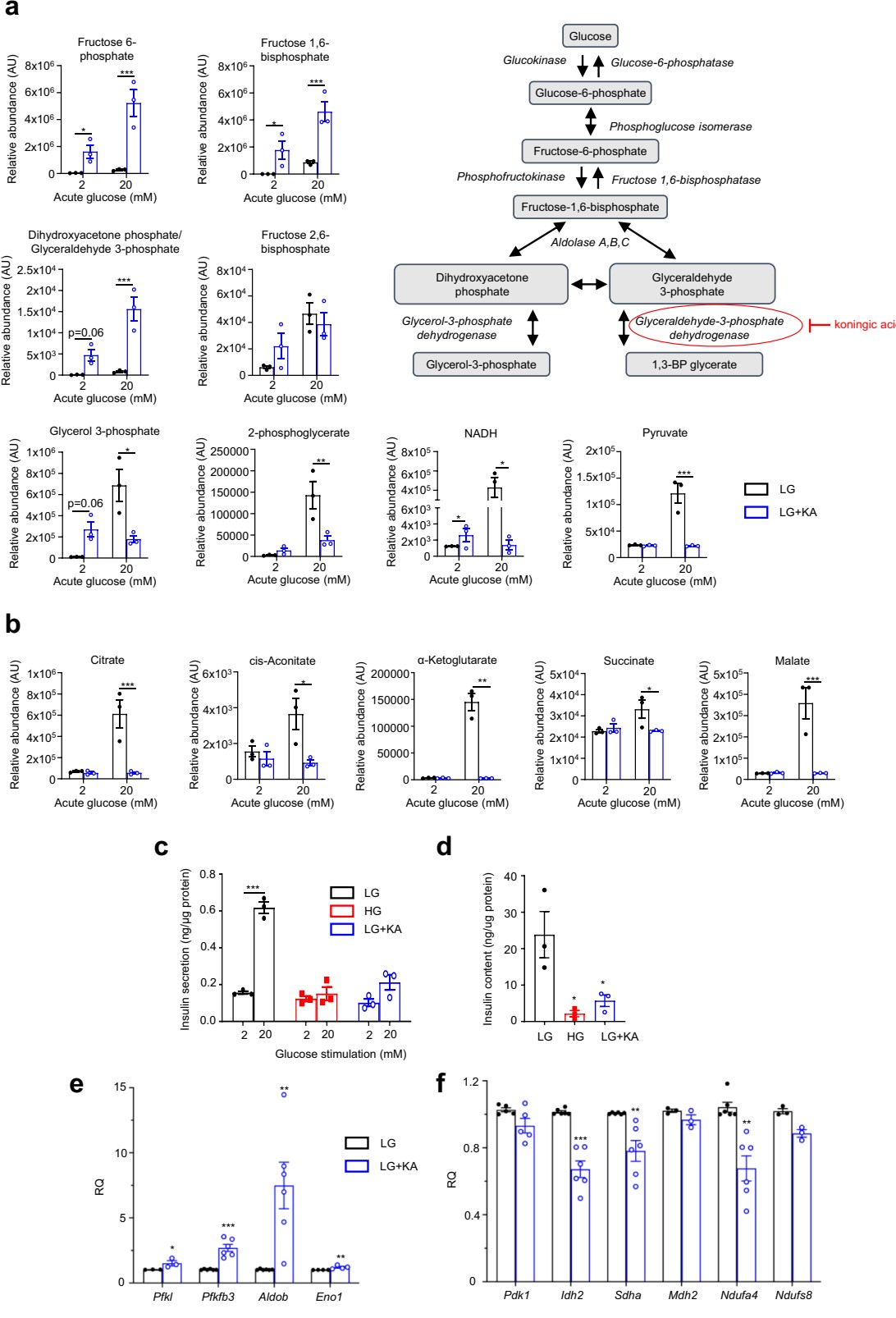

**Fig. 4 | Inhibition of GAPDH induces changes similar to those caused by chronic hyperglycaemia. a** Schematic of glycolysis showing where Koningic acid acts. **a**, **b** Glycolytic (**a**) and TCA cycle (**b**) metabolite abundances in LG-cells cultured for 48 h without (black) or with (blue) 5 μM koningic acid (KA) and subsequently stimulated with 2 mM or 20 mM glucose in the absence of KA (*n* = 3 biologically independent experiments). **c**, **d** Insulin secretion (**c**) and insulin content (**d**) in HG-cells (red) and in LG-cells cultured in the absence (black) or presence (blue) of 5 μM KA for 48 h and then stimulated with 2 mM or 20 mM glucose in the absence of KA

(*n* = 3 biologically independent experiments). **e**, **f** mRNA levels for the indicated glycolytic (**e**) and mitochondrial (**f**) genes assessed by qPCR in LG-cells cultured in the absence (black) or presence (blue) of 5 μM KA. *Pfkl, Mdh2* and *Ndufs8, n* = 3 biologically independent experiments; *Eno1, n* = 4 biologically independent experiments; *Pdk1, n* = 5 biologically independent experiments; *Pfkfb3, Pfkl, Aldob, Idh2, Sdha* and *Ndufa4, n* = 6 biologically independent experiments. All panels show individual data points and mean ± s.e.m. *\**P* < 0.05, *\*\**P* < 0.01, *\*\*\**P* < 0.001, two-tailed unpaired Student's *t* test. Source data are provided as a Source Data file.

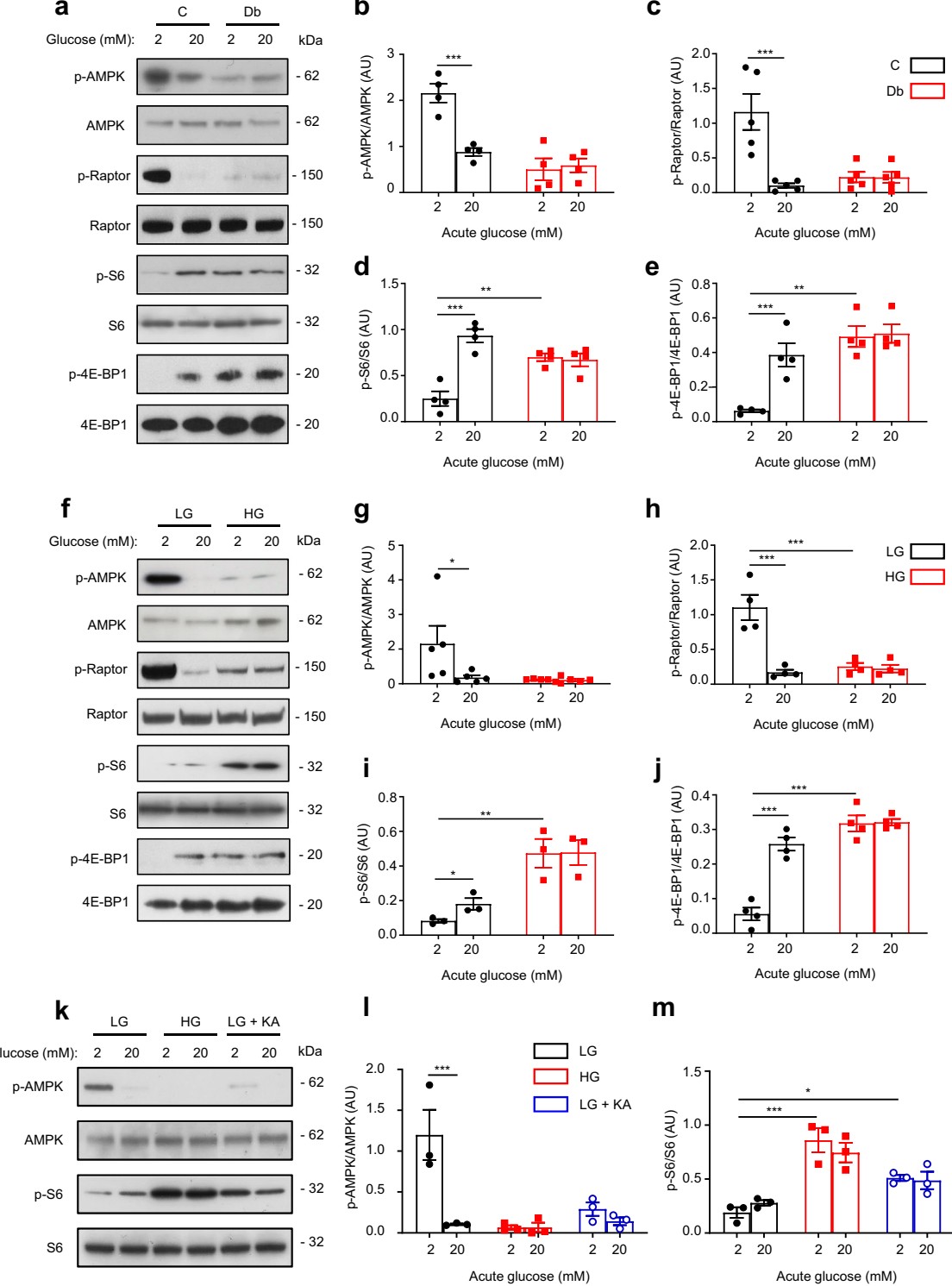

**Fig. 5 | Chronic hyperglycaemia decreases AMPK and increases mTORC1 signalling. a** Representative Western blot of lysates from control (C) and diabetic (Db) islets stimulated with 2 mM or 20 mM glucose for 1 h. Phosphorylated (p) and total AMPK, Raptor, S6 and 4E-BP1. Uncropped blots in Source data. **b–e** Quantitative densitometry analysis of p-AMPK/AMPK (**b**, $n = 4$, 12–15 animals/genotype), p-Raptor/Raptor (**c**, $n = 5$, 14–17 animals/genotype), p-S6/S6 (**d**, $n = 4$, 12–15 animals/genotype) and p-4E-BP1/4E-BP1 (**e**, $n = 4$, 12–15 animals/genotype). Control, black bars. Diabetic, red bars. **f** Representative Western blot of lysates from LG-cells and HG-cells stimulated with 2 mM or 20 mM glucose for 1 h. Phosphorylated (p) and total AMPK, Raptor, S6 and 4E-BP1. Uncropped blots in Source data. **g–j** Quantitative densitometry analysis of p-AMPK/AMPK (**g**, $n = 5$), p-Raptor/Raptor (**h**, $n = 4$

biologically independent experiments), p-S6/S6 (**i**, $n = 3$ biologically independent experiments) and p-4E-BP1/4E-BP1 (**j**, $n = 4$ biologically independent experiments). LG-cells, black bars. HG-cells, red bars. **k** Representative Western blot of lysates from LG-cells, HG-cells and LG-cells cultured for 48 h with 5 μM koningic acid (KA), and then stimulated with 2 or 20 mM glucose for 1 h. Phosphorylated (p) and total AMPK and S6. Uncropped blots in source data. **l, m** Quantitative densitometry analysis of p-AMPK/AMPK (**l**, $n = 3$ biologically independent experiments) and p-S6/S6 (**m**, $n = 3$ biologically independent experiments). LG-cells, black bars. HG-cells, red bars. LG-cells + KA, blue bars. All panels show individual data points plus mean ± s.e.m. *$P < 0.05$, **$P < 0.01$, ***$P < 0.001$, two-tailed unpaired Student's $t$ test. Source data are provided as a Source Data file.

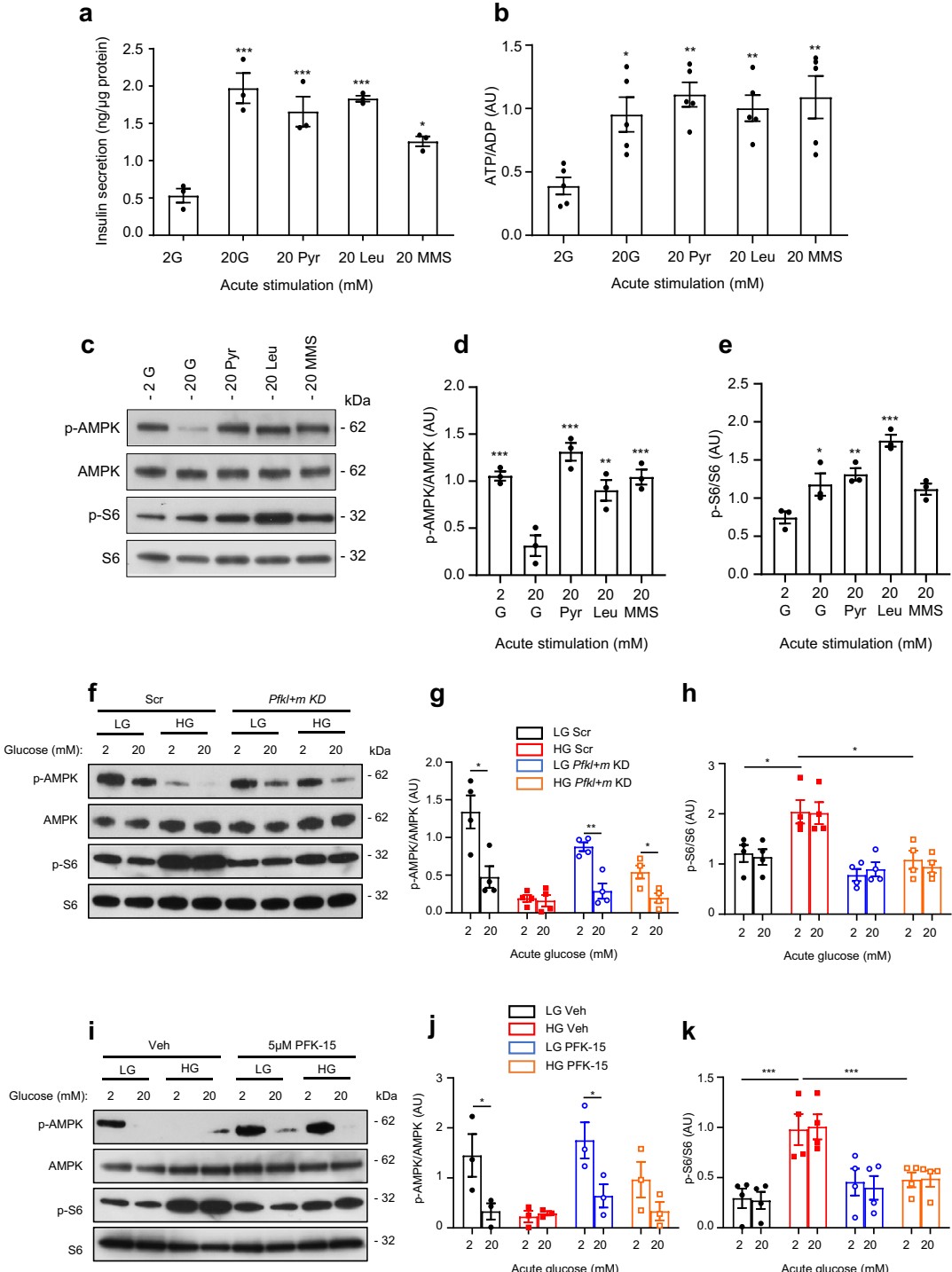

**Fig. 6 | Mitochondrial substrates elevate ATP and insulin secretion but do not suppress AMPK signalling. a, b** Insulin secretion (**a**, *n* = 3 biologically independent experiments) and ATP/ADP ratio (**b**, *n* = 5 biologically independent experiments) in LG-cells stimulated with 20 mM glucose (G), methyl pyruvate (Pyr), leucine (Leu) or monomethylsuccinate (MMS) for 1 h. **c** Representative Western blot of lysates from LG-cells acutely exposed to 2 mM glucose or 20 mM of the indicated mitochondrial substrates for 1 h. Uncropped blots in source data. **d, e** Quantitative densitometry analysis of p-AMPK/AMPK (**d**) and p-S6/S6 (**e**) in LG-cells stimulated with 20 mM glucose (G), methyl pyruvate (Pyr), leucine (leu) or monomethysuccinate (MMS) for 1 h (*n* = 3 biologically independent experiments). **f** Representative Western blot of lysates from cells transfected with scrambled siRNA (Scr) or *Pfkl* and *Pfkm* siRNA (*Pfkl* + *m* KD) and cultured at low (LG) or high (HG) glucose for 48 h. Cells were subsequently stimulated with 2 mM or 20 mM glucose for 1 h. Phosphorylated (p) and total AMPK and S6. Uncropped blots in

source data. **g, h** Quantitative densitometry analysis of p-AMPK/AMPK (**b**, *n* = 4 biologically independent experiments) and p-S6/S6 (**c**, *n* = 4 biologically independent experiments). LG-cells (black), HG-cells (red), LG-cells with *Pfkl*+*m* KD (blue), HG-cells with *Pfkl*+*m* KD (orange). **i** Representative Western blot of lysates from LG- and HG-cells cultured for 48 h in the presence of 0.05% DMSO (veh) or 5 μM of the PFKFB3 inhibitor, PFK-15, and then stimulated with 2 mM or 20 mM glucose for 1 h. Phosphorylated (p) and total AMPK and S6. Uncropped blots in source data. **j, k** Quantitative densitometry analysis of p-AMPK/AMPK (**b**, *n* = 3 biologically independent experiments) and p-S6/S6 (**c**, *n* = 4 biologically independent experiments). LG-cells (black), HG-cells (red), LG-cells + PFK-15 (blue), HG-cells + PFK-15 (orange). Veh, vehicle (0.05% DMSO). Panels show individual data points and mean ± sem. *P < 0.05, **P < 0.01, ***P < 0.001. One-way ANOVA with Bonferroni post hoc test (**a, b, d, e**) and two-tailed unpaired Student's *t* test (**g, h, j, k**). Source data are provided as a Source Data file.

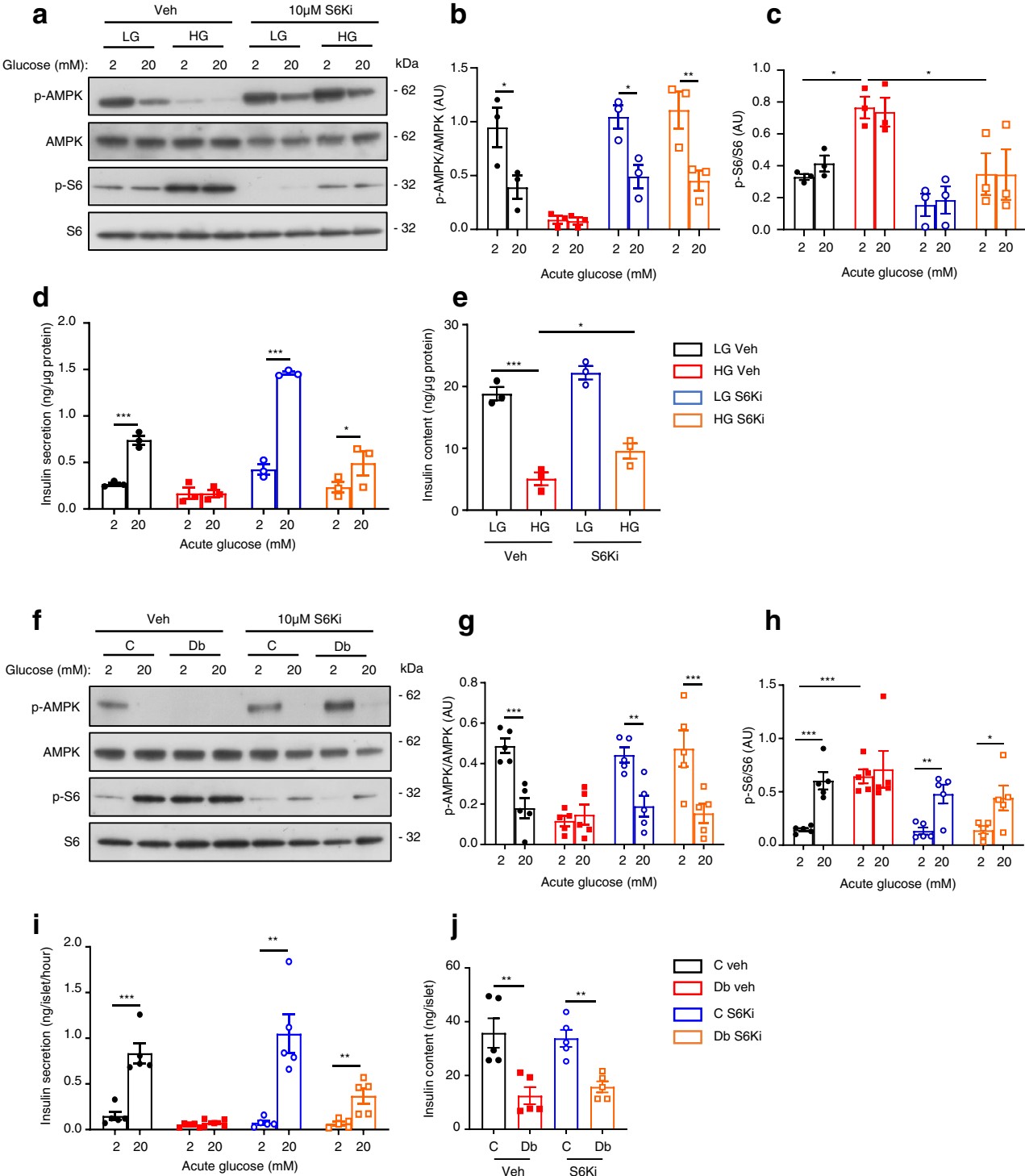

**Fig. 7 | S6-kinase inhibition partially reverses the effects of chronic hypergly-caemia. a** Representative Western blot of lysates from LG- and HG-cells cultured for 48 h ± 10 μM of the S6-kinase inhibitor PF-4708671 (S6Ki), and then stimulated with 2 mM or 20 mM glucose for 1 h in the absence of S6Ki. Phosphorylated (p) and total AMPK and S6. Uncropped blots in source data. **b, c** Quantitative densitometry analysis of p-AMPK/AMPK (**b**, *n* = 3 biologically independent experiments) and p-S6/S6 (**c**, *n* = 3 biologically independent experiments). LG-cells (black), HG-cells (red), LG-cells+S6Ki (blue), HG-cells+S6Ki (orange). **d** Insulin secretion and (**e**) insulin content from LG- and HG-cells cultured for 48 h ± 10 μM S6Ki and stimu-lated with 2 or 20 mM glucose for 30 min in the absence of S6Ki (*n* = 3 biologically independent experiments). **f** Representative Western blot of lysates from control

(C) and diabetic (Db) islets incubated for 48 h ± 10 μM S6Ki and then stimulated with 2 mM or 20 mM glucose for 1 h in the absence of S6Ki. Phosphorylated (p) and total AMPK and S6. Uncropped blots in source data. **g, h** Quantitative densitometry analysis of p-AMPK/AMPK (**g**, *n* = 5, 13–16 animals/genotype) and p-S6/S6 (**h**, *n* = 5, 15–18 animals/genotype). Control islets (black), Diabetic islets (red), Control islets + S6Ki (blue), Diabetic islets + S6Ki (orange). **i** Insulin secretion and **j** insulin content from control and diabetic islets incubated for 48 h ± 10 μM S6Ki and then stimulated with 2 mM or 20 mM glucose for 1 h (*n* = 5 animals/genotype). All panels show individual data points and mean ± s.e.m. *P < 0.05, **P < 0.01, ***P < 0.001, two-tailed unpaired Student's *t* test. Veh, vehicle (0.05% DMSO). Source data are provided as a Source Data file.

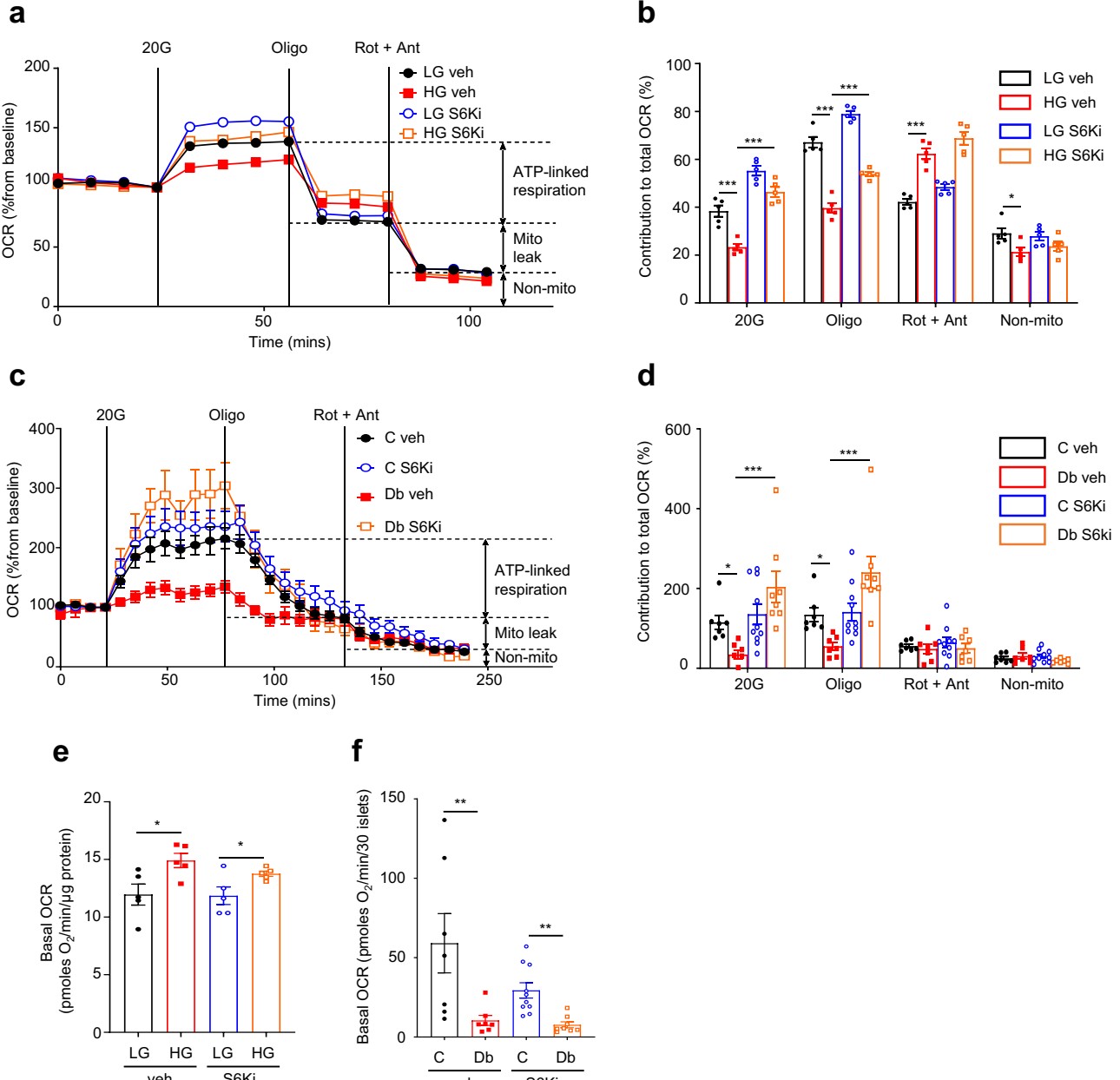

Fig. 8 | **Inhibition of S6 kinase prevents the effects of chronic hyperglycaemia on oxidative metabolism. a** Oxygen-consumption rate (OCR) of LG- and HG-cells cultured for 48 h ± 10 μM S6-kinase inhibitor PF-4708671 (S6Ki). OCR is expressed as the percentage change from the OCR baseline (2 mM glucose) following sequential addition of 20 mM glucose (20 G), 1 μM oligomycin (Oligo) and 0.5 μM rotenone + 0.5 μM antimycin A (Rot + Ant) ($n = 5$ biologically independent experiments per group). LG-cells (black), HG-cells (red), LG-cells+S6Ki (blue), HG-cells +S6Ki (orange). **b** Percentage change in OCR when glucose was raised from 2 mM to 20 mM (20 G, $n = 5$ replicates/group), ATP-linked OCR (Oligo), OCR required to maintain the mitochondrial leak (Rot + Ant) and non-mitochondrial OCR (non-mito) ($n = 5$ replicates/group). Same data as in (**a**). **c** OCR in control and diabetic islets cultured for 48 h ± 10 μM S6Ki. OCR is expressed as the percentage change from the OCR baseline (2 mM glucose) and after sequential addition of 20 mM glucose (20 G), 5 μM oligomycin (Oligo) and 5 μM rotenone + 5 μM antimycin A (Rot + Ant). Control islets (black, $n = 7$ mice examined over seven independent experiments), Diabetic islets (red, $n = 7$ animals examined over seven independent

experiments), Control islets + S6Ki (blue, $n = 7$ mice examined over ten independent experiments), Diabetic islets + S6Ki (orange, $n = 7$ mice examined over 8 independent experiments). **d** Percentage change in OCR of control and diabetic islets when glucose was raised from 2 mM to 20 mM (20 G), ATP-linked OCR (Oligo), OCR required to maintain the mitochondrial leak (Rot + Ant) and non-mitochondrial OCR (non-mito). Same data as in (**c**). **e** Basal OCR at 2 mM glucose in LG- and HG-cells cultured for 48 h ± 10 μM S6Ki ($n = 5$ biologically independent experiments per group). **f** Basal OCR at 2 mM glucose in control and diabetic islets cultured for 48 h ± 10 μM S6Ki Control islets (black, $n = 7$ mice examined over seven independent experiments), Diabetic islets (red, $n = 7$ animals examined over seven independent experiments), Control islets + S6Ki (blue, $n = 7$ mice examined over ten independent experiments), Diabetic islets + S6Ki (orange, $n = 7$ mice examined over 8 independent experiments). All panels show individual data points and mean ± sem. *$P < 0.05$, **$P < 0.01$, ***$P < 0.001$. ± s.e.m. two-tailed unpaired Student's $t$ test. Veh, vehicle (0.05% DMSO). Source data are provided as a Source Data file.

argue that S6K signalling is involved in the regulation of glycolytic, but not mitochondrial, gene expression in response to chronic hyperglycaemia.

### Inhibition of Pdk1 enhances secretion in diabetic islets

Next, we investigated the mechanism by which mTORC1 might influence mitochondrial function, and hence GSIS, in chronic hyperglycaemia. Pyruvate dehydrogenase kinase 1 (PDK1) phosphorylates the E1 subunit of the pyruvate dehydrogenase complex and thereby inhibits PDH activity and pyruvate entry into the TCA cycle[44]. Acute elevation of glucose from 2 to 20 mM increased phosphorylation of PDHe1α in control islets (Fig. 9a, b). By contrast, PDHe1α phosphorylation was enhanced at basal (2 mM) glucose in diabetic islets and unaffected by 20 mM glucose (Fig. 9a, b). Diabetic islets also had increased *Pdk1* mRNA and protein levels[5]. Inhibition of S6K (48 h culture with PF-4708671) reversed the enhanced PDHe1α phosphorylation at 2 mM glucose in diabetic islets (Fig. 9a, b), arguing it is mediated by mTORC1 activation.

As expected if PDH is inhibited, diabetic islets showed impaired mitochondrial metabolism in response to acute stimulation with Me-pyruvate. Compared to control islets, basal NAD(P)H (measured at 2 mM glucose) was increased, and glucose-stimulated NAD(P)H reduced (Supplementary Fig. 7a, c, d), as previously reported[5,45]. Similarly, Me-pyruvate (20 mM) increased NAD(P)H autofluorescence in control islets but the response was impaired in diabetic islets (Supplementary Fig. 7b, c, d). Me-pyruvate also failed to stimulate insulin secretion in HG-cells (Supplementary Fig. 7e), in agreement with the marked inhibition of PDH (Fig. 9a, b), and the reduction in pyruvate carboxylase protein[5]. Insulin secretion in response to Me-succinate or leucine was also reduced (Supplementary Fig. 7e), implying mitochondrial metabolism is also impaired at later steps in the TCA cycle or electron transport chain (ETC), consistent with the downregulation of TCA and ETC proteins.

Acute and chronic hyperglycaemia regulated PDHe1α phosphorylation in LG-cells and HG-cells in a similar manner to that observed in control and diabetic islets (Fig. 9c, d); and chronic GAPDH inhibition in LG-cells recapitulated the effect of chronic hyperglycaemia on PDHe1α phosphorylation (Fig. 9c, d). In addition, Me-pyruvate-stimulated insulin secretion was reduced in HG-cells and in LG-cells cultured chronically with KA (Fig. 9e). Together, these data suggest that chronic hyperglycaemia leads to PDH inhibition via mTORC1 signalling, and that this pathway is triggered via a glycolytic metabolite upstream of GAPDH.

Finally, we examined if inhibiting PDK1 was able to restore insulin secretion in diabetic islets. Figure 9f, g shows that acute application of the universal PDK inhibitor, PS10[46], reduced PDHe1α phosphorylation in response to 20 mM glucose and amplified GSIS in diabetic islets, despite the reduction in insulin content. This confirms that PDH inhibition is partially responsible for the impaired GSIS caused by chronic hyperglycaemia.

## Discussion

Our results show that chronic hyperglycaemia impairs glucose metabolism via increased levels of a glycolytic metabolite lying upstream of GAPDH. Three pieces of evidence support this view: (i) the ability of mannoheptulose to prevent the effects of chronic hyperglycaemia, indicating it is not caused by glucose but by a downstream metabolite; (ii) the failure of chronic pyruvate to recapitulate the effects of chronic hyperglycaemia, indicating the metabolite lies upstream of pyruvate; and (iii) the fact that knockdown or chronic inhibition of GADPH in INS-1 cells cultured at low glucose mimics the effects of chronic hyperglycaemia. Our data further show that this glycolytic metabolite mediates activation of mTORC1 and inhibition of AMPK. The former leads to changes in metabolic gene expression, oxidative phosphorylation and insulin secretion that can be partially reversed by inhibition

of its downstream target S6 kinase. In addition, chronic hyperglycaemia causes inhibition of the metabolic enzymes GAPDH and PDH, which contributes to the suppression of glucose metabolism.

What might be the glycolytic metabolite responsible for altered β-cell metabolism? Several glycolytic intermediates/enzymes are known to serve as sensors that couple metabolism to mTORC1 activity in other cell types. The first is F1,6BP which, in mouse embryonic fibroblasts, binds to aldolase, enabling it to interact with the lysosomal vATPase complex and thereby cause simultaneous activation of mTORC1 and inhibition of AMPK[34,38,39]. This yin-yang relationship controls the balance between anabolic (low glucose) and catabolic (high glucose) processes, with F1,6BP/aldolase serving as the postulated glucose sensor. Secondly, elevation of DHAP activates mTORC1 in HEK cells[35]. Third, glyceraldehyde-3-phosphate (GA3P) prevents GAPDH binding to Rheb and thereby inhibits mTORC1[36]. Finally, the PFK-axis controls glucose-driven mTORC1 activation by E2F1, with PFK and PFKFB3 forming a complex that interacts with RagB GTPase-Ragulator at the lysosome[40]. Metabolomics revealed that F1,6BP and DHAP/GA3P were all elevated in diabetic islets and HG-cells, as well as at low glucose by GAPDH inhibition, suggesting one or more of them may be involved in mediating activation of mTORC1.

In most cells, AMPK activity is increased by elevation of cytosolic AMP when metabolism falls and is inhibited when metabolism is increased[47,48]. As expected, AMPK activity was highest at low glucose and was reduced by acute glucose elevation in control islets and LG-cells. Surprisingly, however, AMPK activity was not inhibited by acute stimulation of LG-cells with mitochondrial substrates, despite their ability to increase the ATP/ADP ratio. Thus the reduction in AMPK activity cannot be due to lower AMP and ADP, the classical regulators of AMPK. This result fits with the idea that AMPK activity in β-cells is regulated by a glycolytic intermediate, such as F1,6BP, which would not be elevated by mitochondrial substrates. In agreement with this idea, chronic exposure of INS-1 cells to pyruvate or leucine did not cause downregulation of AMPK activity and the response to acute glucose elevation was retained.

Strikingly, AMPK activity produced by acute application of low glucose was substantially downregulated in HG-cells and diabetic islets. In both cases, S6K inhibition prevented this effect suggesting it is mediated, directly or indirectly, via mTORC1 activation. A reduction in AMPK activity has also been reported in islets from patients with type 2 diabetes[49]; and in islets from mice with mildly elevated plasma glucose (10 mM $v$. 8 mM) due to high fat feeding[50]. Activation of mTORC1 by acute application of Me-pyruvate, Me-succinate or leucine in LG-cells did not lead to AMPK inhibition. Thus we favour the idea that inhibition of AMPK by mTORC1 activation in chronic hyperglycaemia is mediated indirectly; e.g., via restoration of normal metabolism or interaction with the lysosomal vATPase[38].

Chronic mTORC1 activity led to attenuated mitochondrial glucose metabolism and impaired glucose-stimulated insulin secretion that is mediated, at least in part, by activation of S6K. Hyperglycaemia-induced upregulation of several glycolytic genes was reduced by S6K inhibition, including aldolase B, *Pdk1* and 6-phosphofructo-2-kinase/fructose-2,6-biphosphatase 3 (*Pfkfb3*), a bifunctional enzyme that converts F6P to F2,6BP, a potent activator of phosphofructokinase. Upregulation of *Pfkfb3* may help explain the elevated F2,6BP levels observed in diabetic islets and will contribute to enhanced F1,6BP levels. In addition, the increase in activity of PDK1 produced by chronic hyperglycaemia was reversed by S6K inhibition.

Pyruvate is metabolised in the mitochondria both via conversion to acetyl-CoA by the pyruvate dehydrogenase (PDH) complex and via conversion to oxaloacetate by pyruvate carboxylase (PC). Normally, about equal amounts of pyruvate are metabolised by each route[10,51]. The activity of PDH is inhibited by phosphorylation of its PDHe1α subunit by pyruvate dehydrogenase kinases. Diabetes (islets) or chronic hyperglycaemia (INS-1 cells) markedly enhanced *Pdk1*

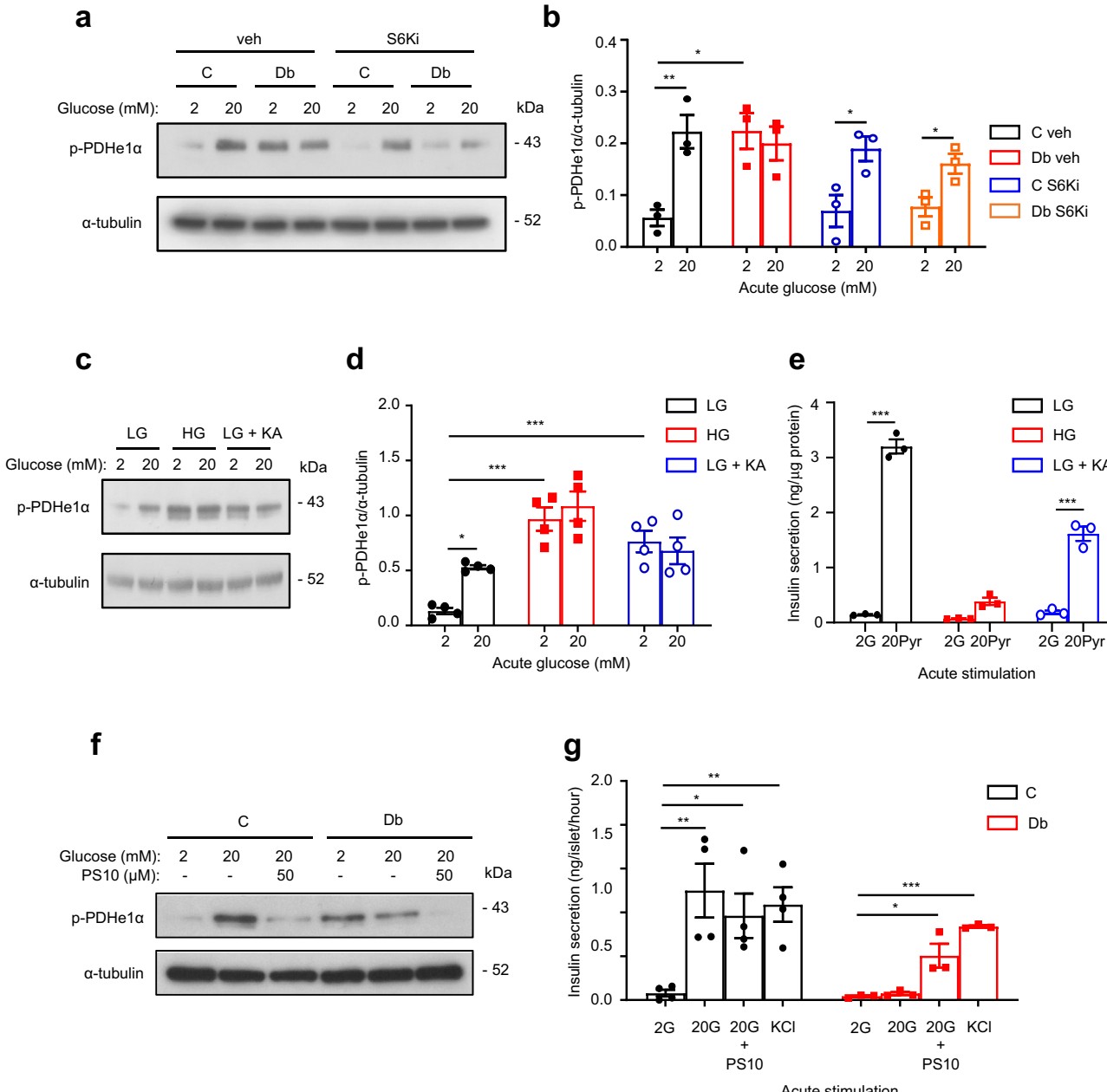

**Fig. 9 | Effects of chronic hyperglycaemia, koningic acid and PS10 on PDH signalling. a** Representative Western blot of lysates from control (C) and diabetic islets (Db) cultured in the presence or absence of 10 μM S6-kinase inhibitor PF-4708671 (S6Ki) and then stimulated with 2 mM or 20 mM glucose for 1 h. Phosphorylated (p) PDHe1α. Loading control, α-tubulin. Uncropped blots in source data. **b** Quantitative densitometry analysis of p-PDHe1α/ α-tubulin ($n = 3$, 11 animals/genotype). Control islets (black), Diabetic islets (red), Control islets + S6Ki (blue), Diabetic islets + S6Ki (orange). Veh, vehicle (0.05% DMSO). **c** Representative Western blot of lysates from LG-cells, HG-cells and LG-cells cultured for 48 h with 5 μM koningic acid (KA), then stimulated with 2 mM or 20 mM glucose for 1 h. Phosphorylated (p) PDHe1α. Loading control, α-tubulin. Uncropped blots in source data. **d** Quantitative densitometry analysis of p-PDHe1α/α-tubulin ($n = 4$

biologically independent experiments). LG-cells (black), HG-cells (red), LG-cells +KA (blue). **e** Insulin secretion from HG-cells, and LG-cells cultured with or without 5 μM koningic acid (KA) for 48 h and subsequently stimulated acutely with 2 mM glucose (G) or 20 mM methyl pyruvate (Pyr) for 30 min ($n = 3$ biologically independent experiments). **f** Representative Western blot of lysates from control (C) and diabetic (Db) islets stimulated with 2 mM or 20 mM glucose ± 50 μM of the PDK inhibitor PS10 for 1 h. Uncropped blots in source data. **g** Insulin secretion from control and diabetic islets stimulated with 2 mM or 20 mM glucose (G), 20 mM glucose + 50 μM PS10, or 20 mM KCl for 1 h. Control islets (black, $n = 4$ animals), Diabetic islets (red, $n = 4$ animals). All panels show individual data points and mean ± s.e.m. *$P < 0.05$, **$P < 0.01$, ***$P < 0.001$, two-tailed unpaired Student's *t* test. Source data are provided as a Source Data file.

expression and phosphorylation of PDHe1α. Similarly, PDK activity was increased and PDH activity reduced in diabetic GK rat islets[52]. Thus this route for pyruvate metabolism is much reduced and acetyl-CoA may become rate-limiting. Pyruvate entry into the TCA cycle via PC is also reduced as PC is downregulated[5]. Together, this will lead to reduced flux through the TCA cycle and may explain why NAD(P)H levels and

glucose oxidation are not increased by glucose stimulation in diabetic islets. The ability of PS10, which inhibits PDK1, to amplify insulin secretion in diabetic islets supports this idea. The failure of Me-pyruvate to elevate NAD(P)H in diabetic islets is consistent with inhibition of PDK1, but is more likely to indicate a failure of the TCA cycle and oxidative phosphorylation, as methyl succinate was also unable to

stimulate insulin secretion in HG-cells and most TCA and ETC proteins were downregulated[5].

GAPDH activity was dramatically reduced in 2-week diabetic islets (this paper) despite a threefold increase in protein abundance[5]. Both these changes were also observed, albeit to a lesser extent, in INS-1 cells exposed to chronic hyperglycaemia. Attenuated GAPDH activity may be due to insufficient cytosolic NAD+[53]. Pancreatic β-cells express very low levels of lactate dehydrogenase, and the mitochondrial glycerol-3-phosphate dehydrogenase (mGPDH) shuttle is the predominant source of NAD+ in the β-cell[31]. Impaired mGPDH shuttle activity has been reported in human islets from T2D donors[54]. Thus the reduced activity of the mGPDH shuttle may contribute to lower GAPDH activity. Another possibility is succination of GAPDH by the Krebs cycle intermediate fumarate, which causes irreversible inactivation of the enzyme and thereby reduces glycolysis[55]. Succination of GAPDH was increased in muscle of diabetic rats[56], and in β-cells from diabetic mice lacking fumarate dehydrogenase in their β-cells[57]. However, fumarate levels were unchanged in our diabetic βV59M islets at 2 mM glucose, and actually reduced at 20 mM glucose (Fig. 3).

Inhibition of GAPDH (in concert with enhanced PFK activity) will contribute to the elevation we observe in upstream metabolites such as F1,6BP and F6P. It may also explain the decrease in glycolysis (measured as the release of tritiated water from labelled glucose at enolase) observed in diabetic islets[5]. Coupled with the marked increase in aldolase B, which favours the formation of F1,6BP (i.e., the back reaction), and other enzymes involved in glycogenesis[5], inhibition of GAPDH may contribute to the marked increase in glycogen found in diabetic islets[58].

With the exception of *Pdk1*, almost all mitochondrial genes we analysed were downregulated in diabetic islets at both the mRNA level and even more potently at the protein level[5]. Similar changes in gene expression were observed in HG-cells and in LG-cells exposed to KA. Mannoheptulose prevented the effects of chronic hyperglycaemia on gene expression, and chronic pyruvate did not recapitulate them. Thus, changes in mitochondrial gene/protein expression must be caused by a glycolytic metabolite upstream of GAPDH. However, at least in INS-1 cells, this metabolite does not regulate mitochondrial gene expression via the mTORC1/S6K signalling pathway, as the mitochondrial gene changes were not prevented by S6K inhibition. This may account for the fact that the ATP-linked OCR in HG-cells was only partially restored by S6K inhibition. S6K inhibition was also unable to reverse the dramatic reduction in basal OCR in diabetic islets, despite restoring glucose-stimulated ATP-linked respiration. Likewise, insulin content appears to be regulated by a separate mechanism as its loss in chronic hyperglycaemia was neither prevented (INS-1 cells) nor restored (islets) by S6K inhibition. Thus mTORC1-S6K signalling may play a partial, but not a sole, role in response to chronic hyperglycaemia in both INS-1 cells and islets.

In conclusion, we propose that inhibition of β-cell metabolism in diabetes is mediated by accumulation of one or more glycolytic metabolites lying between PFK and GAPDH (i.e., F1,6BP, DHAP or GA3P) (Fig. 10). Their accumulation leads to the simultaneous activation of mTORC1 and inhibition of AMPK. mTORC1 activation leads to upregulation of PDK1, which results in reduced entry of pyruvate to the TCA cycle and thus a failure to generate sufficient ATP to support $K_{ATP}$ channel closure and insulin secretion. In addition, GAPDH is profoundly inhibited, impairing glycolytic flux and leading to further accumulation of upstream metabolites, and greater mTORC1 activation. This limits β-cell metabolism in diabetes (Fig. 10).

Most of the changes in metabolic enzyme activity, metabolite levels, mTORC1 and AMPK activity that we see in HG-cells or diabetic islets are also observed in LG-cells or control islets when glucose is acutely elevated from 2 to 20 mM. This suggests a possible mechanism for how acute hyperglycaemia can initiate impaired β-cell mitochondrial metabolism and glucose intolerance. The activity of the ETC and

mitochondrial hyperpolarisation become rate-limiting at high substrate levels[59]. Thus when glucose is acutely elevated in control islets mitochondrial NADH will accumulate[5,45], and consequently, mitochondrial NAD+ levels will fall, slowing the activity of TCA cycle hydrogenases. Elevated mitochondrial NADH is also expected to reduce the activity of the mitochondrial glycerol phosphate and malate/aspartate shuttles (by product inhibition), leading to an increase in *cytosolic* NADH and a concomitant fall in cytosolic NAD+. In turn, this may reduce GAPDH activity enhancing pooling of upstream metabolites. This may account for the activation of mTORC1 and inhibition of AMPK observed in response to acute exposure to 20 mM glucose in LG-cells and control islets. The increase in mTORC1 activity will also enhance activity of PDK1, which will inhibit PDH leading to a gradual reduction in glucose metabolism and ATP synthesis, and causing further pooling of upstream metabolites.

If euglycaemia is quickly restored, these changes will rapidly reverse. However, changes in metabolic gene/protein expression induced by prolonged hyperglycaemia will not be so rapidly reversed and their accumulation may lead to the development of impaired glucose intolerance (IGT). Once IGT is established, we postulate it initiates a vicious spiral in which elevated plasma glucose impairs β-cell metabolism and insulin secretion further, causing greater hyperglycaemia and fuelling the progression of IGT to diabetes. As little as 8 mM glucose appears to be sufficient to initiate this cycle in human islets[60].

Glucose fails to elicit a similar response in most other cell types because of two key differences that are crucial to the role of the β-cell as a metabolic sensor. First, most other cells use hexokinase for glucose phosphorylation, which has a low Km, saturates at <5 mM glucose and is allosterically inhibited by G6P. Consequently, the increase in G6P produced by glucose elevation is limited. Second, most cells possess LDH which hydrolyses cytosolic NADH to NAD+ and thereby regenerates NAD+ for GAPDH activity, and they also possess the lactate transporter MCT1, which enables LDH activity to continue. This maintains glycolysis even when pyruvate metabolism is low.

Our results clearly demonstrate that excess glucose metabolism, rather than excess glucose, causes β-cell failure. Reducing glucose metabolism might therefore be an effective strategy to prevent the progressive decline in β-cell function that occurs in diabetes. The ability of mannoheptulose to prevent the effects of chronic hyperglycaemia suggests partial inhibition of glucokinase might be a viable strategy. Indeed, there is accumulating evidence that partial glucokinase inhibition both in vitro and in vivo can help preserve β-cell function and mass in mouse models of diabetes[61–63]. Although at first sight, it may seem paradoxical to suggest reducing GCK activity may be therapeutic in T2D, evidence from people with a heterozygous inactivating mutation in GCK provides support for this idea. Despite mild fasting hyperglycaemia (~6.5 mM), these patients require no medication, their hyperglycaemia does not progress and their prevalence of diabetic complications is not increased[64,65]. Glucose homoeostasis is simply maintained at a higher set point, resulting in mild asymptomatic hyperglycaemia. The counter-regulatory response to hypoglycaemia is also improved[66]. This suggests that even as much as 50% loss of GCK activity is not harmful. However, the aim of any therapy would be to restore GCK activity in diabetes to that found at normal blood glucose levels in control β-cells (and no further). Glucokinase is expressed in very few tissues (β-cells, liver, some neurones and endocrine cells). In all these cells, partial inhibition of glucokinase, leading to a reduction in G6P in the face of excess glucose, is likely to be beneficial.

## Methods

### Animal experiments

All animal studies were conducted in accordance with the UK Animals (Scientific Procedures) Act (1986) and approved by the local Department of Physiology Anatomy and Genetics (University of Oxford) ethical review committee. βV59M mice (which hemizygously express

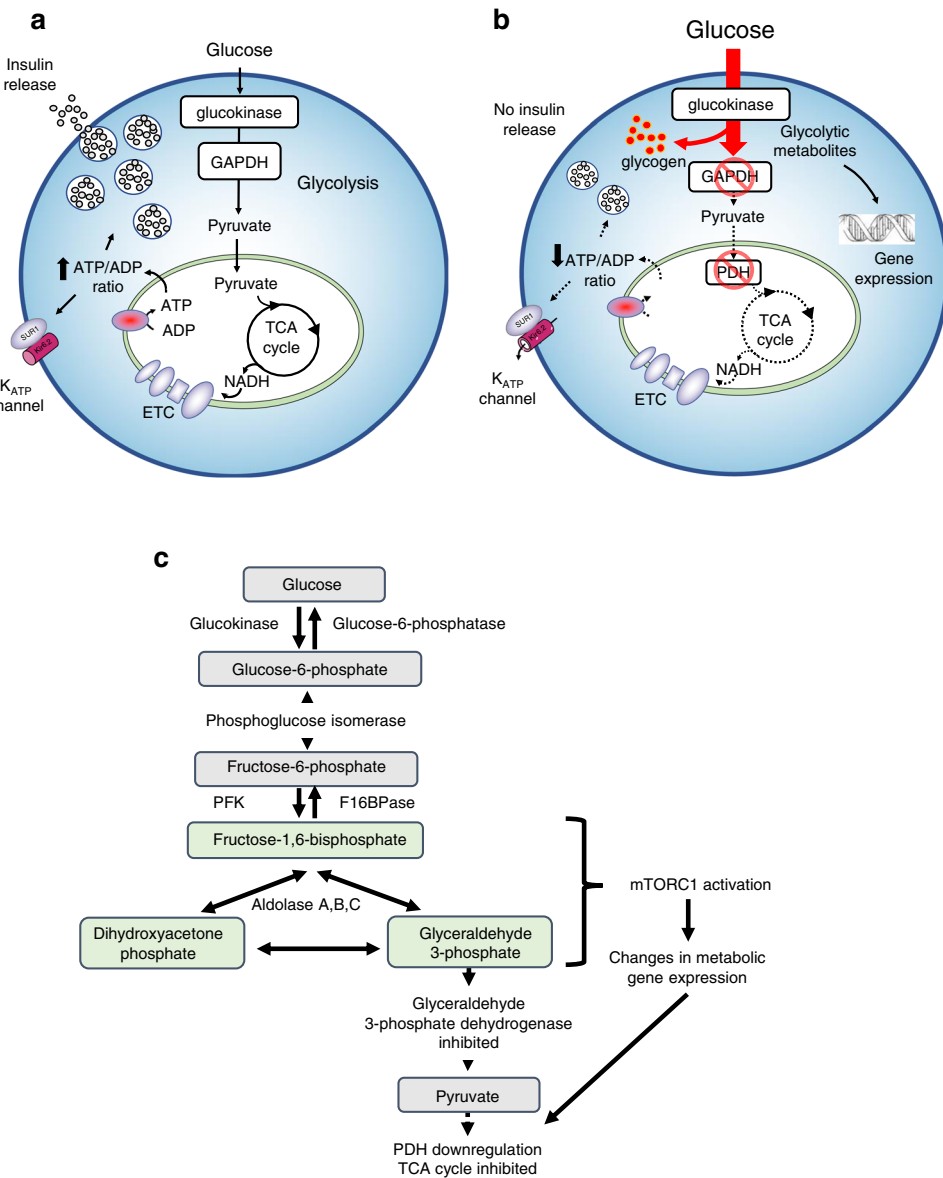

**Fig. 10 | Effects of chronic hyperglycaemia on β-cell metabolism. a, b** Cartoons illustrating metabolism in normoglycaemic/non-diabetic (**a**) and chronic hyperglycaemic/diabetic (**b**) conditions. **a** In non-diabetic cells glucose is metabolised via glycolysis and oxidative phosphorylation to produce an increase in the ATP/ADP ratio which closes $K_{ATP}$ channels and initiates electrical activity, calcium influx and insulin secretion. **b** In diabetic β-cells glucose uptake is markedly increased due to the elevated extracellular glucose concentration. However, the activity of GAPDH and PDH are inhibited leading to the accumulation of metabolites upstream of GAPDH which cause changes in gene expression that downregulate PDH and GAPDH activity further. Because metabolism is reduced, glucose-dependent changes in the ATP/ADP ratio, $K_{ATP}$ channel activity and insulin secretion are impaired. The enhanced activity of glucokinase also leads to a build-up of G6P which is channelled into glycogen. **c** Schematic of glycolysis showing the metabolic steps (highlighted) implicated in mTORC1 activation and consequent changes in metabolic gene expression. Metabolic steps that are inhibited are also indicated.

the inducible *Kir6.2-V59M* transgene specifically in their β-cells) were generated using a Cre-lox approach[30]. The background strain was C57BL6/J (Jackson Labs, USA). Both male and female mice were used. Transgene expression was induced at 12 weeks of age by subcutaneous injection of tamoxifen (Sigma, 10 μl/g body weight of 20 mg/ml) in corn oil. Mice were sacrificed 2 weeks later, when blood glucose levels had risen to >20 mM. Tamoxifen-injected wild-type, RIPII-Cre-ER and floxed STOP Kir6.2-V59M littermates (pooled) were used as controls. All mice were bred in-house from established stocks. Mice had unrestricted access to water and a regular chow diet (63% carbohydrate, 23% protein, 4% fat; Special Diet Services, RM3). They were maintained on a 12 h light-dark cycle at 21 °C and 45–55% relative humidity. Body weight and blood glucose levels were monitored routinely. Blood glucose levels were measured from the tail vein using the Freestyle Lite device and Freestyle Lite test strips (both Abbott).

**Islet isolation and culture.** Mice were killed by cervical dislocation, and islets were isolated essentially as described[67]. In brief, pancreata were inflated with collagenase solution at 1 mg/ml (Collagenase NB 8 Broad Range, Nordmark, S1745602), resected and placed in a water bath at 37 °C for 10 min. After washing, islets were purified on a Histopaque-1119 (Merck, #11191) / Histopaque-1083 (Merck, #10831) gradient. Islets were cultured overnight (or for 48 h, as indicated) in RPMI-1640 medium containing 11 mM glucose (control) or 25 mM glucose (diabetic), plus foetal bovine serum (10% (v/v), penicillin (100 U/ml) and streptomycin (0.1 mg/ml) solution (all Thermo-Fischer-Scientific) at 37 °C, in a humidified atmosphere of 5%$CO_2$/95% air. The

glucose concentration of RPMI was chosen to reflect the randomly fed blood glucose of the animals.

## INS-1 (832/13) cell culture

INS-1 (832/13) cells (abbreviated here as INS-1 cells) were originally developed by Claes Wollheim (Geneva). They were cultured in RPMI-1640 medium supplemented with 10% FBS, 1% Pen/Strep, 50 μM β-mercapto-ethanol, 1 mM Na-pyruvate, 10 mM HEPES, and 2 mM glutamine (standard culture medium; all Sigma-Aldrich) in a humidified atmosphere of 5% $CO_2$/95% air at 37 °C. Unless otherwise stated, the glucose concentration was 11 mM. In order to replicate normoglycaemic (control) and hyperglycaemic (diabetic) conditions in vitro, INS-1 cells were subsequently cultured at either 5 mM (low glucose: LG-cells) or 25 mM glucose (high glucose: HG-cells) for 48 h before the experiment. For inhibitor studies, cells were cultured at 5 or 25 mM glucose for 48 h either with the indicated inhibitor or with the appropriate vehicle control for 48 h. In some experiments, cells were cultured in 20 mM methyl pyruvate or leucine instead of glucose (and glucose was omitted from the culture medium).

## Insulin secretion

*INS-1 cells:* INS-1 (832/13) cells were cultured in RPMI-1640 medium containing 5 or 25 mM glucose for 48 h. On the day of the assay, cells were washed twice in Krebs–Ringer-bicarbonate buffer containing (in mM): 140 NaCl, 3.KCl, 0.5 $NaH_2PO_4$, 2 $NaHCO_3$ [saturated with $CO_2$], 1.5 $CaCl_2$, 0.5 $MgSO_4$, 10 HEPES (pH 7.4) and 0.1% (w/v) fatty acid-free (FFA) BSA. Cells were pre-stimulated with 2 mM glucose Krebs buffer at 37 °C for 60 min, after which the buffer was removed, and cells were incubated with Krebs buffer containing 2 or 20 mM glucose for 30 min. The supernatant was removed and cells harvested either in acid ethanol (for total insulin content) or in RIPA lysis buffer (for protein content). Insulin levels in the supernatant and cell lysates were determined by insulin ELISA (Mercodia, Uppsala, Sweden). Insulin secretion and insulin content were normalised to the protein content of the well.

*Islets:* Insulin secretion was assayed in triplicate on size-matched islets, isolated from control and diabetic β-V59M mice (2 weeks after diabetes induction). Islets were pre-stimulated for 1 h in Krebs–Ringer bicarbonate buffer + 0.1% (w/v) FFA-BSA containing 2 mM glucose, and then stimulated for 1 h (10 islets/well) at 37 °C in Krebs–Ringer solution with either 2 mM glucose, 20 mM glucose or 20 mM KCl. Islets were harvested in acid ethanol to determine insulin content. Secreted and total insulin content were quantified by ELISA (Mercodia, Uppsala, Sweden). Data were expressed per islet.

**ATP measurements.** INS-1 cells were cultured at 5 mM glucose and then incubated with glucose or mitochondrial substrates as indicated for 1 h. ATP/ADP ratio was measured using a bioluminescent ADP/ATP ratio assay kit (Abcam, ab65313) according to the manufacturer's instructions.

## Respirometry

The Seahorse XF24 Extracellular Flux Analyser (Seahorse Bioscience, Copenhagen, Denmark) was used to assess a range of metabolic parameters by real-time monitoring of cellular oxygen-consumption rate (OCR), as described by Haythorne et al (2019)[5]. In brief, INS-1 cells were cultured at 5 or 25 mM glucose for 48 h and washed in serum-free unbuffered assay medium (DMEM 5030, Sigma) containing 2 mM glucose for 1 h prior to measurement. Isolated islets were incubated overnight in RPMI supplemented with either 11 mM glucose (control islets) or 25 mM glucose (diabetic islets). Islets were seeded at 30 islets/well in XF 24-well islet capture microplates in unbuffered DMEM containing 2 mM glucose and 0.1% fatty acid-free BSA for 1–2 h prior to measurement.

Glucose-stimulated respiration was measured by the addition of 20 mM glucose. Mitochondrial efficiency was assessed using compounds that inhibit specific mitochondrial processes: ATP-linked respiration (oligomycin), and proton leak (antimycin A + rotenone). Data are presented as either pmol $O_2$/min/30 islets or were normalised to the 4th baseline measurement (100%). The % change in OCR following the addition of a compound/substrate was also calculated.

## SDS-PAGE and western blotting

INS-1 (832/13) cells were cultured in RPMI medium at 5 or 25 mM glucose for 48 h. They were then serum-starved for 1 h in HEPES-buffered saline containing (mM) 135 NaCl, 5 KCl, 1 $MgCl_2$, 1 $CaCl_2$, 10 HEPES (pH 7.4 with NaOH), plus 5 or 25 mM glucose, followed by stimulation with 2 or 20 mM glucose (in saline) for 1 h. Cells were harvested in RIPA buffer (Sigma-Aldrich) containing phosphoprotease inhibitor (Roche) and protease inhibitor (Roche). Islets (-200) from control or 2-week diabetic β-V59M mice were serum-starved and stimulated for 60 min, similar to INS-1 (832/13) cells, and lysed in ice-cold RIPA buffer (Sigma-Aldrich) containing protease and phosphatase inhibitors.

Protein concentration was determined using a Pierce BCA protein assay kit (Thermo Fisher Scientific). Protein isolation and immunoblotting procedures were as described previously[68]. Briefly, 10–20 μg of protein lysates were subjected to SDS-PAGE and electrotransferred to a nitrocellulose membrane, and immunoreactive proteins were identified by chemiluminescence. Primary and secondary antibodies used are listed in Supplementary Table 1A. Gel bands were quantified by densitometry using Image Studio Lite software (Licor). mTORC1 signalling was determined by the ratio of phosphorylated (p-S6) and total ribosomal protein S6 (S6). Phosphorylated PDHe1α was normalised to α-tubulin due to lack of a suitable total PDHe1α antibody. α-tubulin was used as a loading control.

Uncropped blots are available in the Source data file and the Supplementary information.

**qPCR.** Total RNA was isolated using Qiazol lysis reagent or the RNeasy Mini Kit (both Qiagen) according to the manufacturer's instructions. RNA concentration was determined using a NanoDrop ND-1000 spectrophotometer (Thermo Scientific) and RNA reverse transcribed. Quantitative PCR was performed using TaqMan probes (Supplementary Table 1B) and the Applied Biosystems StepOne Plus Real-Time PCR system (Applied Biosystems). All reactions were performed in duplicate. Data were quantified according to the delta-delta Ct method with normalisation to the housekeeping gene *HSPA8* or *HPRT1*.

**siRNA experiments.** INS-1 832/13 cells were transfected using Lipofectamine RNAiMAX (Life Technologies, Paisley, UK). The following siRNAs were used: ON-TARGETplus siRNA SMARTpool for rat *Pfkl*, ON-TARGETplus siRNA SMARTpool for rat *Pfkm*, ON-TARGETplus siRNA SMARTpool for rat *Aldoa*, ON-TARGETplus siRNA SMARTpool for rat *Aldob*, ON-TARGETplus siRNA SMARTpool for rat *Gapdh* and ON-TARGETplus Non-Targeting Control siRNA (Horizon Discovery Biosciences Limited, Cambridge, UK). For dual gene knockdowns, cells were transfected at 60% confluency with *Pfkl* and *Pfkm* siRNA, or *Aldoa* and *Aldob* siRNA, at a final concentration of 50 nM of each siRNA, or with control siRNA (non-targeting siRNA) at a final concentration of 100 nM. For *Gapdh* knockdown experiments, cells were transfected with *Gapdh* siRNA or control siRNA (non-targeting siRNA) at a final concentration of 50 nM. For all experiments, cells were cultured at 5 mM glucose (LG) or 25 mM glucose (HG) for 48 h, after which cells were lysed to extract total RNA or protein.

**Enzyme activities.** Enzyme activities were measured using commercial assay kits according to the manufacturer's instructions: hexokinase (Abcam, ab136957), aldolase (Abcam, ab196994), phosphofructokinase (Abcam, ab155898), glyceraldehyde-3-phosphate dehydrogenase (Abcam, ab204732) and fructose-1,6-bisphosphatase (Biovision, K590).

**Metabolomics using anion-exchange chromatography–mass spectrometry.** To measure metabolite abundances under conditions at which insulin secretion takes place, 50 islets from control and diabetic mice were preincubated for 1 h in 2 mM glucose Krebs–Ringer bicarbonate buffer + 0.1% (w/v) FFA-BSA, followed by incubation for 1 h with either 2 mM or 20 mM glucose, at 37 °C. Metabolites were extracted from the islets in ice-cold 80% methanol$_{(aq)}$, using 1 µl of solvent per islet. Samples were vortex mixed for 5 min, centrifuged at $2.2 \times 10^4 \times g$ for 20 min and the supernatant was transferred into autosampler vials to remove cell debris. Samples were normalised to DNA content (INS-1 cells) or islet number (islets). Analysis of samples was performed using anion-chromatography coupled directly to high-resolution orbitrap mass spectrometry[69]. A 5 µl partial loop injection was used for all analyses and chromatographic separation was performed using a Dionex ICS-5000+ Capillary HPIC system (Thermo Scientific, San Jose, CA, USA) coupled to a Q Exactive hybrid quadrupole-Orbitrap mass spectrometer (Thermo Scientific, San Jose, CA, USA). A Dionex IonPac AS11-HC column (2 × 250 mm, 4 µm; Thermo Scientific, San Jose, CA, USA) at 30 °C was used with an aqueous hydroxide ion gradient as follows: 0.0 min, 0.0 mM; 1.0 min, 0.0 mM; 15.0 min, 60.0 mM; 25.0 min, 100.0 mM; 30.0 min, 100.0 mM; 30.1 min, 0.0 mM; and 37.0 min, 0.0 mM. Online conductive ion suppression was achieved using a Dionex AERS 500e 2 mm (Thermo Scientific, San Jose, CA, USA) in external water mode. The flow rate was 0.250 ml/min and the total run time was 37 min. The Q Exactive mass spectrometer was equipped with a HESI II probe in negative ion mode with source parameters set as follows: sheath gas flow rate, 70; auxiliary gas flow rate, 20; sweep gas flow rate, 0; spray voltage, 3.6 kV; capillary temperature, 320 °C; S-lens RF level, 70; and heater temperature, 350 °C. MS and MS$^2$ scan parameters were set as follows: microscans, 2; resolution, $7 \times 10^4$; AGC target, $1 \times 10^6$ ions; maximum IT, 120 ms; and scan range, 60–900 $m/z$. MS/MS scan parameters were set as follows: microscans, 2; resolution, $1.75 \times 10^4$; AGC target, $1 \times 10^5$ ions; maximum IT, 250 ms; loop count, 10; multiplex count, 1; isolation window, 2.0 $m/z$; collision energy, 35; minimum AGC target, $5 \times 10^3$ ions; apex trigger 1–15 s; charge exclusion, 3–8, >8; and dynamic exclusion, 20.0 s. Metabolomics data was processed and analysed using Xcalibur v.4.2 (Thermofisher Scientific, Waltham, USA), Progenesis QI v.2.0 (Waters Corp, Manchester, UK) and MetaboAnalyst 5.0 (www.metaboanalyst.ca).

DHAP was not identified in all metabolomics samples and thus was measured in islets using a fluorometric assay. Briefly, islets were incubated in 2 mM glucose Krebs buffer for 1-h before being exposed to 2 or 20 mM glucose for a further hour. Islets were then collected and processed according to the manufacturer's instructions (Abcam ab197003). Data are normalised to islet number.

**Imaging.** Endogenous NAD(P)H was imaged using a Zeiss Axio-Zoom.V16 microscope at 10 to 14-fold magnification. Mouse islets isolated from control and 2-week diabetic βV59M mice were preincubated for 90 min at room temperature in extracellular solution (ECS) containing (in mM) 140 NaCl, 4.6 KCl, 2.6 CaCl$_2$, 1.2 MgCl$_2$, 1 NaH$_2$PO$_4$, 5 NaHCO$_3$ and 10 HEPES, (pH 7.4, with NaOH) and 2 mM glucose. Groups of islets then were positioned in an imaging chamber placed on a heated stage (34 °C) and perifused continuously with ECS (rate 60 µl/min) at 34 °C. NAD(P)H was excited at 365 nm and the emission collected at 445 nm. Time-lapse images were collected every 60 s. Images were analysed using open-source Fiji (ImageJ1.53 f) software (http://fiji.sc/Fiji), with the whole islets taken as regions of interest (ROI). The time-lapse series numerical data were analysed using Igor-Pro package (Wavemetrics).

### Statistical analysis
Unless otherwise stated, results are presented as mean ± s.e.m. Unless otherwise stated, for islet experiments, $n$ indicates the number of mice and for INS1-cell studies, $n$ indicates the number of experiments. Most INS-1-cell experiments had 2 or more (usually 3) technical replicates and the mean value of all replicates was taken as $n = 1$. For oxygen-consumption experiments, $n$ indicates the number of replicates (wells). For Western blot experiments using islets, islets from multiple mice had to be pooled in order to obtain a sufficient amount of protein so we state both the number of experiments ($n$) and the total number of mice used.

Significance was tested using Student's $t$ test, one-way or two-way ANOVA as indicated in the figure legends, using Graphpad Prism software. Post-test corrections are indicated in the legends. Differences were considered statistically significant if $P < 0.05$ using FDR corrections for multiple testing where applicable.

### Reporting summary
Further information on research design is available in the Nature Research Reporting Summary linked to this article.

## Data availability
The raw mass spectrometry data files from which the results were generated for Fig. 3a, b, Fig. 4a, b, and Supplementary Fig. 3a, b have been deposited in the Oxford Research Archive, and are available at https://ora.ox.ac.uk/objects/uuid:1859eafb-6fff-4ae6-ba61-f4d224e0ae51 with the https://doi.org/10.5287/bodleian:dmOmXAVJ5 (resolving to https://doi.org/10.5287/bodleian:dmOmXAVJ5). The processed data are provided in Supplementary Data files 1 and 2. The authors declare that all data supporting the findings of this study are available within the paper, its Supplementary Information files, its Supplementary Data files, or the Source Data file. Supplementary Information accompanies this paper. Source data are provided with this paper.

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

## Acknowledgements
We thank Dr. Steve Ashcroft for valuable discussions and Dr. Thomas Hill for technical help. We thank the animal house staff for animal care. We thank the UK Medical Research Council (MR/T002107/1 to F.M.A., P.R. and E.H.), the Biotechnology and Biological Sciences Research Council (BB/R017220/1 to F.M.A. and BB/R013829/1 to J.M.), the John Fell Fund (006657 to E.H.) and the Nuffield Benefaction for Medicine/Wellcome Institutional Strategic Support Fund (Oxford MSIF grant 0007293 to E.H.) for support.

## Author contributions
E.H. carried out the respirometry, enzyme activity and insulin-secretion experiments, and E.H., J.S. and M.C. carried out the Western blotting. M.L., I.P., R.T.-E. and M.R. performed and analysed the qPCR experiments, and J.M., J.W.-T., F.M.A. and E.H. performed metabolomics experiments and data analysis. E.H., A.I.T., G.S. and P.R. carried out imaging experiments. R.T.-E. and M.C. performed glucokinase assays; F.M.A. supervised the project. E.H. and F.M.A. designed experiments, analysed the data and co-wrote the paper. All authors revised and approved the manuscript.

## Competing interests
The authors declare no competing interests.

## Additional information

**Correspondence and requests** for materials should be addressed to Elizabeth Haythorne or Frances M. Ashcroft.

