## [Peer Review File · Nature Communications]

Altered glycolysis triggers impaired mitochondrial metabolism and mTORC1 activation in diabetic β -cellsREVIEWER COMMENTS

Reviewer #1 (Remarks to the Author):

Haythorne et al studied the mechanisms leading to mitochondrial dysfunction and impaired insulin secretion under conditions of chronic hyperglycemia. To explore the effects of chronic hyperglycemia on β -cell metabolism the authors used two models: 1. the insulin-secreting cell line INS-1 832/13, cultured at 25mM glucose for 48h, and 2. islets of diabetic β V59M mice, a model of human neonatal diabetes induced by an activating mutation in the gene encoding the KATP channel. They show that one or more glycolytic metabolites downstream of glucokinase and upstream of GAPDH (possibly F1,6BP or DHAP) mediate the effects of chronic hyperglycemia. This putative metabolite induced marked upregulation of mTORC1 along with downregulation of AMPK. Increased mTORC1 activity caused inhibition of pyruvate dehydrogenase (PDH), which in turn inhibited pyruvate entry into the tricarboxylic acid cycle resulting in inhibition of oxidative phosphorylation and insulin secretion. In addition, diabetes and prolonged exposure of INS-1 cells to high glucose (HG) markedly inhibited the glycolytic enzyme GAPDH, thereby impairing glucose metabolism. They suggest that in diabetes, alterations in glycolysis dysregulate mTORC1-AMPK signaling with subsequent inhibition of glucose oxidation and insulin secretion.

Overall, the findings are interesting, however, part of the findings are not clear and needs to be explained. Furthermore, there are important issues that should be clarified. First, it is not clear which metabolite(s) upstream of GAPDH account for the stimulation of mTORC1 and/or inhibition of AMPK. Second, it is not clear how sustained activation of mTORC1 in diabetes affects AMPK and PDH activities. Third, the data that mTORC1 (S6K) inhibition is sufficient to restore beta-cell function are not convincing and are mainly based on a single pharmacological inhibitor of S6K. See specific comments below:

Major critique

1. Which glycolytic metabolite(s) stimulate mTORC1? Metabolomics revealed upregulation of glycolytic intermediates (F1,6BP and DHAP) as well as fructose and pentose pathway intermediates. These alterations were accompanied by increased PFK, FBP, and aldolase activities and decreased GAPDH activity. Identifying the metabolic intermediate(s) that activate mTORC1 in beta-cells is difficult considering the multiple changes that occur in diabetes. I suggest that the authors perform knockdown experiments of key enzymes that were altered in their analysis, i.e. GAPDH, PFK, aldolase B, and FBP in INS1 cells cultured at high and low glucose, and clarify which metabolite(s) stimulate mTORC1 in beta-cells.

2. The finding that GAPDH activity is inhibited in diabetes and in INS1 cells cultured at HG is interesting. There were contradictory effects of HG on GAPDH protein level and activity. This should be explained. A recent multi-omics analysis of human islets derived from patients with IGT or T2D showed alterations in mitochondrial gene expression and in aldolase B (Wigger et al, Nature Metabolism 2021), which is consistent with the current manuscript. Aldolase B expression was increased in human islets exposed to

HG; however, changes in GAPDH gene and/or protein expression were not reported. Are these changes unique to the models used in this study? The relevance to human islets should be confirmed by testing the effects of HG on GAPDH protein level and activity.

3. How does mTORC1 activation inhibit AMPK? It is commonly believed that mTORC1 is inhibited by AMPK. The authors suggest that AMPK is regulated by mTORC1. Any mechanistic explanation for this observation? In addition, part of the experimental data is inconsistent with this conclusion. In Fig. 6c, methyl-pyruvate, leucine, and mono-methyl succinate stimulated in parallel mTORC1 (pS6) and AMPK. This argues against the suggestion that mTORC1 inhibits AMPK. The finding that 48h incubation of INS1 cells with leucine failed to stimulate mTORC1 at both LG and HG is puzzling (Supplemental Fig 5d-f). Leucine stimulated mTORC1 in the short term. How do the authors explain that the effect is lost during prolonged incubation?

4. There are some concerns regarding the models being used. INS1 cells might be particularly susceptible to glucotoxicity. Part of the key findings should be confirmed in rodent or human islets incubated at HG. In β V59M mice, not all beta-cells express it, and "pure" hyperglycemia hardly reflects T2D. Also, ionic perturbations in beta-cells may influence glucose metabolism independent of hyperglycemia effects.

5. There was a marked reduction of insulin content in both INS1 cells incubated at HG and in diabetic islets. The reduction in content in diabetic islets was not prevented by mannoheptulose or S6Ki. What is the explanation for the reduction in content? Is insulin synthesis impaired? Increased apoptosis? The authors should also express secretion as percentage of content so the reader can appreciate whether the effects on secretion are secondary to reduction in content. It is difficult to interpret the effects on secretion in presence of dramatic effects on content.

6. The authors should add a control experiment in which they test the effects of HG with diazoxide to prevent the stimulatory effects of HG on secretion. Can this correct the impaired GSIS and reduction in content? Does chronic exposure to HG induce the metabolic changes described when continuous stimulation of insulin secretion is prevented?

7. The authors suggest that impaired insulin secretion is due to reduced delivery of glycolytic products (pyruvate) to the TCA cycle. To confirm the absence of mitochondrial disturbances independent of substrate availability, they should check whether the response of diabetic islets and INS1 cells incubated at HG to mitochondrial fuels, e.g succinate ester or leucine, is preserved.

8. It is not clear how sustained activation of mTORC1 induces metabolic reprogramming in beta-cells. S6K inhibition reduced the expression of several metabolic genes, particularly those of glycolysis. How

does S6 kinase regulate the expression of glycolytic genes? The authors should provide a mechanistic explanation for this observation. How does S6 kinase modulate PDK1 activity and PDHe1² phosphorylation? Is there physical interaction between S6 kinase and PDK1 and/or PDH? Can it directly phosphorylate these enzymes? At this stage, the conclusion that mTORC1/S6K induces metabolic reprogramming in diabetic islets is not proven.

9. S6K inhibition at LG markedly augmented GSIS (Fig. 7d) suggesting that its effects on insulin secretion are mTORC1-independent. The experiments testing the effects of mTORC1 inhibition on glucose metabolism and insulin secretion are all based on a single pharmacological inhibitor of S6K. The authors should consider performing genetic manipulations of mTORC1 downstream targets, mainly S6K, to strengthen their interpretation of the findings.

10. Conditional KO of Tsc1/Tsc2 leading to constitutive activation of mTORC1 in beta-cells, hence mimicking the effects of diabetes, results in increased insulin content and secretion and improved glucose tolerance. This phenotype is preserved for a long period of time although late decline in beta-cell function is possible. How do the authors explain that activation of mTORC1 in the diabetic milieu rapidly impairs beta-cell function? These contradictory effects should be explained.

Minor comments

1. Insulin secretion data shown in Supplemental Fig. 4c should be presented in a uniform manner as in the rest of the manuscript.

2. In the Discussion, the authors state that: "reducing GK activity may be therapeutic in T2D, evidence from people with a heterozygous inactivating mutation in GK provides support for this idea". Patients with MODY2 develop mild hyperglycemia and therefore rarely develop complications. This doesn't imply that moderate inhibition of glucokinase can ameliorate diabetes. I suggest toning down this statement.

Reviewer #2 (Remarks to the Author):

This is an impressive study that advances our understanding of the abnormal insulin secretion found in the diabetic state. Because of the thorough approach to the problem and the well-chosen measurement tools that were used, a variety of surprising findings emerged, which do not provide all of the answers but move us to a new level of understanding. Clarity of the thought processes and interpretations help the reader through complicated metabolic systems.

One of many strengths of the study was the use of two good beta cell model systems, INS-1 cells and diabetic betaV59M mice that have an inducible KATP channel mutation. The first important surprise was that the methyl ester form of pyruvate did not mimic the changes seen with high glucose concentrations. This helped focus attention on something important going on at the GAPDH level. The metabolomics data provided the key findings that there were large increases in F6P, F1, 6BP and F26BP but down-regulation of most TCA intermediates. This fit with the decrease in the activity of GAPDH and PDH, implicating the potential importance of some glycolytic metabolites downstream of glucokinase and upstream of GAPDH; that is, between G6P and pyruvate. It was particularly interesting that there was a marked decrease in the activity of GAPDH, despite a 3-fold increase in the GAPDH protein. This combination of findings led to the hypothesis that one or more of these metabolites caused upregulation of mTORC1 and downregulation of AMPK. Data were then generated that strengthened this hypothesis.

The conclusions are well-supported by the data provided, and the discussion was well-written.

Specific issues:

1. It would be helpful for readers to have a cartoon or some kind of diagram to show the key interactions that seem to be involved.
2. Not much attention was paid to beta cell death, which is OK because it was not studied. However, some mention might be helpful. Looking carefully at the beta cell mass data of Rahier and Henquin (Rahier J. Pancreatic alpha cell mass in European subjects with type 2 diabetes. *Diabetologia*. 2011;54(7):1720) It seems that the rate of beta cell death is very slow in spite of decades of hyperglycemia.
3. The suggestion in the discussion that inhibition of glucokinase might be helpful when people have hyperglycemia does not seem to make sense. People with GK mutations do very well for two reasons. First, their glucose elevations are mild so they are largely spared from the glucose -induced complications. Second, they have good GSIS, which probably helps smooth out their control. With a GK inhibitor it seems glucose levels would climb and increase the glycemic effects on complications.
4. Throughout the paper, data are often described as n=3, sometimes as “in duplicate” and sometimes replicates are mentioned. Anyway, there is room for improvement. Please provide more information about how experiments with multiple replicates are handled.

5. Supp Fig. 1 Panels c: and d: Expressing secretion as ng/ml/islet/hour does not help us to understand what we really need to know, which is how much insulin is secreted per islet per hour.

6. Supp Fig 2. There may be a better way to show these data. On my version, there are no asterix designating statistical significance. Also, there is no arrow pointing at F16BP.

7. Supp Fig 3. There must be some way to better highlight the important changes. I wonder if excel files would be helpful.

8. Supp Fig 4. It would be helpful to change the patterns and/or colors so that there is better discrimination between the vertical bars.

Reviewer #3 (Remarks to the Author):

Haythorne and colleagues examine the mechanisms by which chronic hyperglycemia and type II diabetes lead to reduced β -cell function and whether these pathological alterations are associated with mTORC1 activation. The authors claim that glycolytic metabolites downstream of hexokinase and upstream of GAPDH mediate the effects of diabetes and chronic hyperglycemia on β -cell metabolism. This happens through hyperactivation of mTORC1, which leads to alterations in insulin secretion and oxidative phosphorylation. In addition, the authors show that hyperglycemia inhibits GAPDH and pyruvate dehydrogenase (PDH) activities. The authors use control or Db islets, and INS1 cells cultured in low (LG) or high glucose (HG) to recapitulate diabetes and chronic hyperglycemia. Overall, the authors support the idea that gradual damage of β -cell metabolism, induced by increasing hyperglycemia, promotes diabetes development and suggest that reducing glycolysis at the level of hexokinase may slow this progression.

General comments:

While the concept proposed by the authors is interesting, the mechanisms by which glycolysis controls chronic hyperglycemia remain unidentified. The study is preliminary but provides a hint on how potential glycolytic intermediates upstream of GAPDH and downstream of hexokinase support chronic hyperglycemia and diabetic state. The take-home message of the paper is confusing and needs to be further elucidated. The authors should focus on clearly identifying the identity of the metabolite or metabolic enzyme involved in controlling insulin secretion and insulin content in INS1 cells or diabetic islets.

The data with the S6K1 inhibitor (Fig. 7d,e) are not convincing. Treatment with S6K1 inhibitor slightly increases insulin content in HG cells. This suggests that the contribution of S6K in the control of hyperglycemia is relatively negligible. The authors should use rapamycin (mTORC1 inhibitor) or even Akt inhibitor since Akt has also been proposed to respond to glucose metabolism.

A previous study has linked dihydroxyacetone phosphate (DHAP) to mTORC1 activation. So, does the supplementation of dihydroxyacetone decrease insulin content and secretion in LG cells? Of course, to get this rescue to work, the authors should express triose kinase (TKFC) in INS1 cells to make sure that exogenous dihydroxyacetone can get converted into DHAP to signal mTORC1 signaling.

Based on the metabolomics data presented in this manuscript, it is hard to pinpoint the specific metabolite responsible for the diabetic phenotype. DHAP and glyceraldehyde 3-phosphate (GA3P) have the exact same mass. Therefore, it is surprising that the authors could differentiate them by classic LC-MS. For example, I am not convinced that GA3P reads presented in Fig. 4a are not a mixture of DHAP/GA-3P. How did the authors ensure that GA3P is not DHAP+GA3P? Moreover, if koningic acid (KA) inhibits GAPDH activity, should not we observe an accumulation of GA3P levels upon LG+KA (20 mM Glucose) condition?

In addition to koningic acid, other inhibitors of glucose metabolism could be used to methodically study the contribution of key glycolytic branches in the control of hyperglycemia and diabetes. Since intermediates of the pentose phosphate pathway are increased upon high glucose condition, targeting G6PD (6-aminonicotinamide) to decrease 6-phosphogluconate and transketolase (oxythiamine) to decrease sedoheptulose 7-phosphate could be employed to assess the effects of these inhibitors on the insulin secretion and content in LG cells.

Specific comments:

The authors should reduce the number of main Figures (The manuscript contains now nine main figures, and the figure could be condensed and presented more succinctly. For example, figure 2, which shows negative data, could be moved to the supplemental item.

AMPK phosphorylation does not necessarily reflect AMPK activity. Therefore, in addition to p-AMPK, the authors should present phosphorylation of AMPK substrates (p-ACC (S79)/ACC or p-RAPTOR (S792)/RAPTOR).

Fig. 5d: P-S6/S6 signal should be improved.

Fig. 6c: The p-S6 signal does not appear to be increased upon 20 mM glucose compared to 2 mM glucose. The authors should provide a better representative western blot.

Other key markers of mTORC1 signaling should be presented, such as p70S6 kinase and 4E-BP1 phosphorylation.

REPLY TO REVIEWERS: Haythorne et al. NCOMMS-21-48720

We thank the reviewers for their careful critique of our paper and their valuable comments. We have addressed these below and believe that the manuscript is now much improved.

Reviewer #1 (Remarks to the Author):

Haythorne et al studied the mechanisms leading to mitochondrial dysfunction and impaired insulin secretion under conditions of chronic hyperglycemia. To explore the effects of chronic hyperglycemia on β -cell metabolism the authors used two models: 1. the insulin-secreting cell line INS-1 832/13, cultured at 25mM glucose for 48h, and 2. islets of diabetic β V59M mice, a model of human neonatal diabetes induced by an activating mutation in the gene encoding the KATP channel. They show that one or more glycolytic metabolites downstream of glucokinase and upstream of GAPDH (possibly F1,6BP or DHAP) mediate the effects of chronic hyperglycemia. This putative metabolite induced marked upregulation of mTORC1 along with downregulation of AMPK. Increased mTORC1 activity caused inhibition of pyruvate dehydrogenase (PDH), which in turn inhibited pyruvate entry into the tricarboxylic acid cycle resulting in inhibition of oxidative phosphorylation and insulin secretion. In addition, diabetes and prolonged exposure of INS-1 cells to high glucose (HG) markedly inhibited the glycolytic enzyme GAPDH, thereby impairing glucose metabolism. They suggest that in diabetes, alterations in glycolysis dysregulate mTORC1-AMPK signaling with subsequent inhibition of glucose oxidation and insulin secretion.

Overall, the findings are interesting, however, part of the findings are not clear and needs to be explained. Furthermore, there are important issues that should be clarified. First, it is not clear which metabolite(s) upstream of GAPDH account for the stimulation of mTORC1 and/or inhibition of AMPK. Second, it is not clear how sustained activation of mTORC1 in diabetes affects AMPK and PDH activities. Third, the data that mTORC1 (S6K) inhibition is sufficient to restore beta-cell function are not convincing and are mainly based on a single pharmacological inhibitor of S6K. See specific comments below:

Major critique

1. Which glycolytic metabolite(s) stimulate mTORC1? Metabolomics revealed upregulation of glycolytic intermediates (F1,6BP and DHAP) as well as fructose and pentose pathway intermediates. These alterations were accompanied by increased PFK, FBP, and aldolase activities and decreased GAPDH activity. Identifying the metabolic intermediate(s) that activate mTORC1 in beta-cells is difficult considering the multiple changes that occur in diabetes. I suggest that the authors perform knockdown experiments of key enzymes that were altered in their analysis, i.e. GAPDH, PFK, aldolase B, and FBP in INS1 cells cultured at high and low glucose, and clarify which metabolite(s) stimulate mTORC1 in beta-cells.

We thank the reviewer for this suggestion. We have now knocked down these genes in INS1 cells cultured at high and low glucose and measured p-S6 and p-AMPK by Western blotting. There are two isoforms of PFK (Pfk1 and Pfkm) expressed at significant levels in beta-cells, and two isoforms of aldolase (AldoA and AldoB). We initially knocked down Pfk1 and AldoB (which are upregulated in diabetes)

independently. These data are included for the reviewer (Reviewer Figs 1 and 2). We also looked at the combined effect of Pfkf and Pfkf, and of AldoA and AldoB, knockdown. These data have been added to the paper.

We now say

“Our data support the idea that a glycolytic metabolite lying between glucokinase and GAPDH mediates the effect of chronic hyperglycaemia on both mTORC1 and AMPK. We reasoned that knockdown of an enzyme upstream of the key metabolite would prevent mTORC1 activation and AMPK inhibition in HG-cells, whereas knockdown of a downstream enzyme might cause their reciprocal activation/inhibition in LG-cells. As F1,6BP^[35], DHAP^[35], and GA3P^[37], as well as PFK itself^[41], have all been implicated in regulating mTORC1 activity, we next knocked down PFK in LG- and HG-cells (Supplementary Fig.4e). The combined knockdown of Pfkf and Pfkf prevented the phosphorylation of S6 and the reduction in AMPK phosphorylation in HG cells (Fig.6f-h). We also tested the effect of chronic PFK-15, an inhibitor of PFKFB3 (6-phosphofructose-2-kinase or PFK2) and thereby an indirect inhibitor of PFK^[42]. Chronic PFK-15 recapitulated the effects of PFK knockdown in HG-cells (Fig.6i-k). These data indicate that the metabolite that mediates the reciprocal activation of mTORC1 and inhibition of AMPK in chronic hyperglycaemia lies between PFK and GAPDH (i.e. F1,6BP, GA3P or DHAP).

Expression of aldolase B is strikingly upregulated in HG cells (Fig.1f), in β V59M diabetic islets^[5], and in islets from T2D donors^[17]. Although we were unable to completely knockdown aldolase B, we found that partial knockdown of both aldolase A+B led to partial upregulation of AMPK activity (Supplementary Fig.4c-f). This is consistent with a role for aldolase/F1,6BP in AMPK regulation. However we did not see a reciprocal inhibition of S6 kinase (Supplementary Fig.4g-i). This suggests a different mechanism may be involved in mTORC1 regulation.”

We also include a Figure for the reviewer, showing that we were only able to knockdown aldolase activity by about 50% (see Reviewer Fig. 2).

Knockdown of FBP1 in LG-cells did not recapitulate the effects of high glucose culture in terms of p-S6 activation (See Reviewer Fig. 3). This is to be expected as there is little gluconeogenesis in beta-cells under euglycaemic conditions.

In addition, we explored the effect of inhibiting entry into the pentose phosphate pathway on insulin secretion and gene expression, as suggested by Reviewer 3. We found that chronic culture with 100 μ M 6-aminonicotinamide (6-AN), which inhibits 6-phosphogluconate dehydrogenase, had little or no effect on the changes in insulin secretion and glycolytic gene expression induced by chronic hyperglycaemia, thus excluding a mechanistic role for pentose phosphate pathway intermediates. Chronic 6-AN attenuated insulin secretion in LG-cells which suggests the pentose phosphate pathway activity is required for normal insulin secretion, as published previously (Peter Spégel et al *Biochem J*, 2013). These data have also now been added to the paper (Supplementary Fig.2).

As the reviewer states it is difficult to define precisely the metabolite that activates mTORC1. This is because (in other cell types) several glycolytic enzymes and their substrates/products participate in regulating mTORC1 in response to glucose (Li et al., *Cell Research* 2021, 31:478–481). These include FBP, DHAP, GA3P and the enzymes PFK, aldolase and GAPDH. Similarly, it is likely that one or more of these metabolites (and enzymes), regulates mTORC1 activity in β -cells. We have now made this clearer in the text.

2. The finding that GAPDH activity is inhibited in diabetes and in INS1 cells cultured at HG is interesting. There were contradictory effects of HG on GAPDH protein level and activity. This should be explained. A recent multi-omics analysis of human islets derived from patients with IGT or T2D showed alterations in mitochondrial gene expression and in aldolase B (Wigger et al, Nature Metabolism 2021), which is consistent with the current manuscript. Aldolase B expression was increased in human islets exposed to HG; however, changes in GAPDH gene and/or protein expression were not reported. Are these changes unique to the models used in this study? The relevance to human islets should be confirmed by testing the effects of HG on GAPDH protein level and activity.

It is not especially surprising that chronic hyperglycaemia reduces GAPDH activity without affecting protein expression. The activity of many enzymes is regulated by post-translational modifications or substrate/co-factor availability. In the case of GAPDH, it is likely that insufficient coenzyme NAD⁺ contributes to the low activity of the enzyme in diabetes, as β -cells express negligible levels of lactate dehydrogenase (which replenishes cytosolic NAD⁺). With regard to human islets, GAPDH mRNA is commonly used as a reference gene for qRT-PCR, which implies its expression does not change in type 2 diabetes (eg. Marselli et al, *Cell Reports* 33,108466). It is also worth noting that GAPDH activity is reduced in the muscle (*Ann N Y Acad Sci.* 2008; 1126, 272–275) and retina of diabetic rats (*Diabetes* 2009; 58, 227–234).

We have also now included a reference to the Wigger paper and added a comment about the marked elevation of AldoB in diabetes. We now say: “AldoB is strikingly upregulated in HG cells (Fig.1), in β V59M diabetic islets [5], and in islets from T2D donors [17].”

Measuring GAPDH activity in human islets has many caveats. In particular, it is far from clear that normal metabolism is retained in islets isolated from cadaver organ donors, which may have been exposed to numerous drugs prior to death. Furthermore, culture of islets is often associated with necrosis in the islet centre which makes analysis of metabolites and enzyme activity problematic. Indeed, 24h culture of rodent islets at high glucose in vitro can actually *potentiate* insulin content and secretion, contrary to what is found in vivo (Rebelato, *Sci Rep.* 8, 13061, 2018). Why this is the case is unclear.

3. How does mTORC1 activation inhibit AMPK? It is commonly believed that mTORC1 is inhibited by AMPK. The authors suggest that AMPK is regulated by mTORC1. Any mechanistic explanation for this observation? In addition, part of the experimental data is inconsistent with this conclusion. In Fig. 6c, methyl-pyruvate, leucine, and mono-

methyl succinate stimulated in parallel mTORC1 (pS6) and AMPK. This argues against the suggestion that mTORC1 inhibits AMPK.

mTORC1 has actually been shown to inhibit AMPK through phosphorylation at AMPK α 1Ser347/ α 2Ser345 (Ling NXY et al *Nat Metab* 1, 41-49, 2020). We examined phosphorylation at this site and found it was phosphorylated at 20mM glucose in LG-cells and at both 2mM and 20mM glucose in HG-cells (Reviewer Figure 4a). This would be consistent with mTORC1 phosphorylation and inhibition of AMPK. This phosphorylation was not prevented by S6 kinase inhibition (Reviewer Figure 4b).

However, mTORC1 may also cause inhibition of AMPK indirectly. For example, as inhibition of mTORC1 signalling via S6 kinase inhibition restores some aspects of metabolism (ie. PDH activity) this may have a knock-on effect that indirectly restores AMPK activity. A second possibility is that mTORC1 and AMPK are reciprocally activated/inactivated via interaction with the lysosomal vATPase (Li et al., *Cell Research* 31:478–481, 2021). This is now clarified in the text. Fig.6c shows acute stimulation with MeP, Me-succinate and leucine in low glucose-cultured INS1 cells, and (as the reviewer states) supports the view that the effect on AMPK is indirect. This is also now made clearer in the text.

The finding that 48h incubation of INS1 cells with leucine failed to stimulate mTORC1 at both LG and HG is puzzling (Supplemental Fig 5d-f). Leucine stimulated mTORC1 in the short term. How do the authors explain that the effect is lost during prolonged incubation?

Leucine activates mTORC1 via a distinct mechanism involving disruption of the Sestrin2-GATOR2 interaction (Wolfson et al., *Science* 351:43-8, 2016). This mechanism has not been shown to be shared by glucose metabolism. The difference between the short-term and long-term effects of leucine may be the result of the amino acid mediating its downstream actions via more than one mechanism, which have different long term outcomes. For example, it is possible that it has a direct stimulatory effect on mTORC1 activity in the short-term, but that chronic exposure leads to long-term changes in gene expression that have the opposite effect. However, the effects of leucine are outside the scope of this paper which is focused on the effects of chronic glucose on β -cell metabolism.

4. There are some concerns regarding the models being used. INS1 cells might be particularly susceptible to glucotoxicity. Part of the key findings should be confirmed in rodent or human islets incubated at HG. In β V59M mice, not all beta-cells express it, and "pure" hyperglycemia hardly reflects T2D. Also, ionic perturbations in beta-cells may influence glucose metabolism independent of hyperglycemia effects.

We point out that, in contrast to this reviewer, reviewer 2 considers our choice of model systems to be 'one of many strengths of the study'. It is true that not all β -cells in the β V59M mice will express the mutation (although most of them do). However, all cells will be exposed to the chronic hyperglycaemia. Furthermore, we have unpublished single cell RNAseq data that shows the same changes in metabolic gene expression in β -cells that do not express the Kir6.2-V59M mutation as in those that do (e.g upregulation of AldoB, etc). We attach a confidential figure for the reviewer (Reviewer Figure 5): these data will be published in a separate paper.

We previously showed that K_{ATP} channel activation *per se* does not cause the β -cell changes we observe as they can be prevented by normalization of glycaemia with insulin therapy (Brereton et al., 2014). Thus we do not believe that ionic perturbations in β -cells (due to the K_{ATP} channel overactivity) will influence glucose metabolism independently of hyperglycaemia effects. This is also now explained in the text.

We now say: The β -cell changes found in diabetic $\beta V59M$ mice are prevented by restoration of euglycaemia with insulin, indicating they are due to hyperglycaemia/hypoinsulinaemia not K_{ATP} channel activation *per se*^[28].

With regard to the comment that pure hyperglycaemia does not reflect T2D, we point out that our aim is to dissect the contribution of hyperglycaemia to β -cell failure – hence our focus on models of ‘pure’ hyperglycaemia. It is also worth noting that the deleterious effects of lipids alone on β -cells are relatively small. As Weir and Bonner-Weir conclude in an insightful review on this topic ‘evidence that lipotoxicity contributes to β -cell secretory dysfunction or death in human diabetes or animal models is weak to non-existent’ (*Ann NY Acad Sci* 1281,92-105, 2013). Furthermore, in contrast to high glucose, chronic exposure to a lipid cocktail alone did not suppress insulin secretion from human islet microtissues; lipids were only effective at suppressing insulin secretion/content when accompanied by elevated glucose (Mir-Coll et al *Int. J. Mol Sci* 22, 1813, 2021).

Finally, there are caveats when using human or rodent islets chronically cultured at high glucose for metabolic experiments. The principal one is the lack of an intact vasculature which means the centre of the islet can become anoxic, which will compromise oxidative metabolism. While this is not a problem for imaging experiments, where surface cells can be selected, it is an issue for metabolic experiments. Human islets also have the problem that they come from cadaver organ donors who may have been exposed to numerous drugs prior to death.

5. There was a marked reduction of insulin content in both INS1 cells incubated at HG and in diabetic islets. The reduction in content in diabetic islets was not prevented by mannoheptulose or S6Ki. What is the explanation for the reduction in content? Is insulin synthesis impaired? Increased apoptosis? The authors should also express secretion as percentage of content so the reader can appreciate whether the effects on secretion are secondary to reduction in content. It is difficult to interpret the effects on secretion in presence of dramatic effects on content.

The reduction in insulin content was *prevented* by mannoheptulose (in INS1 cells, Fig.1c) but was not *reversed* when diabetic islets were isolated and then cultured at high glucose in the presence of mannoheptulose for 48hr (Supplementary Fig.1d). It is possible that the latter is because it takes more time to increase insulin protein expression back to normal levels. We did not treat diabetic mice with mannoheptulose so as yet we cannot say if the reduction in insulin content is *prevented* by mannoheptulose. However, we refer the reviewer to a couple of very recent papers that show genetic reduction of glucokinase activity prevents the loss of insulin content in mice *in vivo* (Yan et al, *Diabetes* 2022; Omori et al, *Diabetes* 2021).

We now say: “Insulin content was not restored by mannoheptulose, suggesting that the drug can prevent the fall in insulin content (in INS1 cells) but may not reverse it (in islets) on the time scale of our experiments.”

We agree that insulin content is not affected by S6Ki, which suggests that although a glucose metabolite mediates the effect of hyperglycaemia on insulin content it does so via a different signalling pathway. We have now made this clearer in the text.

We now say: “Likewise, insulin content appears to be regulated by a separate mechanism as its loss in chronic hyperglycaemia was neither prevented (INS-1 cells) nor restored (islets) by S6K inhibition.”

We have shown previously that hyperglycaemia markedly reduces insulin gene expression in both INS1 cells and diabetic islets, indicating reduced insulin synthesis (Brereton et al., 2014). There is some apoptosis in diabetic β V59M islets, but this is very low, and the reduced insulin content of islets is due to loss of insulin granules and not loss of β -cells (Brereton et al., 2014). Precisely how hyperglycaemia affects insulin gene expression is the subject of a different paper: this paper focuses on the effects of chronic hyperglycaemia on β -cell metabolism.

We prefer to present the data as insulin content and insulin secretion as this involves less data manipulation and the reader can see clearly what is happening to both insulin secretion and content. Some journals now formally request this. Furthermore, even if secretion is expressed as a percentage of content this does not provide a true reflection of what is happening to glucose-stimulated release. There is not necessarily a linear relationship between insulin content and release – for example, content may be low but if most granules are docked/primed then release could be little affected.

6. The authors should add a control experiment in which they test the effects of HG with diazoxide to prevent the stimulatory effects of HG on secretion. Can this correct the impaired GSIS and reduction in content? Does chronic exposure to HG induce the metabolic changes described when continuous stimulation of insulin secretion is prevented?

INS1 cells are able to depolarise and release insulin when they are transferred from low to high glucose; thus until chronic hyperglycaemia causes β -cell failure, they are still able to release insulin. In contrast, β -cells in the β V59M mice are permanently hyperpolarised by the activating K_{ATP} channel mutation and unable to release insulin: their β -cells resemble β -cells treated with diazoxide. Data from β V59M islets show that chronic exposure to high glucose induces the metabolic changes described even when continuous stimulation of insulin secretion is prevented (Brereton et al, 2014). We also previously showed that K_{ATP} channel activation *per se* does not cause the β -cell changes we observe, as they can be prevented by normalization of glycaemia with insulin therapy (Brereton et al., 2014). The latter information has now been added to the paper.

We now say: “The β -cell changes found in diabetic β V59M mice are prevented by restoration of euglycaemia with insulin, indicating they are due to hyperglycaemia /hypoinsulinaemia not K_{ATP} channel activation *per se* (Brereton et al, 2022).”

7. The authors suggest that impaired insulin secretion is due to reduced delivery of glycolytic products (pyruvate) to the TCA cycle. To confirm the absence of mitochondrial disturbances independent of substrate availability, they should check whether the response of diabetic islets and INS1 cells incubated at HG to mitochondrial fuels, e.g succinate ester or leucine, is preserved.

This is a good point. We have indeed confirmed that mitochondrial activity is impaired, independent of substrate availability, in INS1 cells exposed to chronic hyperglycaemia. We found insulin secretion in HG cells in response to acute methyl pyruvate, monomethyl succinate, and leucine was impaired (Supplementary Fig.7e). This is perhaps not surprising given the downregulation of electron transport proteins in diabetes (Haythorne et al., 2019). This data is now added to the paper.

We now say: “Me-pyruvate also failed to stimulate insulin secretion in HG-cells (Supplementary Fig.7e), in agreement with the marked inhibition of PDH (Fig.9a,b), and the reduction in pyruvate carboxylase protein^[6]. Insulin secretion in response to Me-succinate or leucine was also reduced (Supplementary Fig.7e), implying mitochondrial metabolism is also impaired at later steps in the TCA cycle or electron transport chain (ETC), consistent with the downregulation of TCA and ETC proteins”.

8. It is not clear how sustained activation of mTORC1 induces metabolic reprogramming in beta-cells. S6K inhibition reduced the expression of several metabolic genes, particularly those of glycolysis. How does S6 kinase regulate the expression of glycolytic genes? The authors should provide a mechanistic explanation for this observation. How does S6 kinase modulate PDK1 activity and PDHe1 phosphorylation? Is there physical interaction between S6 kinase and PDK1 and/or PDH? Can it directly phosphorylate these enzymes? At this stage, the conclusion that mTORC1/S6K induces metabolic reprogramming in diabetic islets is not proven.

We agree with the reviewer that details of the mechanism by which S6 kinase upregulates expression of glycolytic genes and PDK1 is of interest. However, this is a major topic and will be the subject of a further paper as the current manuscript is already over-long. Previous studies have shown that mTORC1 activation in MEF cells enhances transcription of almost every enzyme in the glycolytic and pentose phosphate pathway, as well as genes involved in fatty acid synthesis (Düvel et al. *Mol Cell* 39, 171-183, 2010). These authors identified cMyc, SREBP and HIF1 α as transcription factors upregulated by mTORC1. They further showed that S6K is the major downstream target of mTORC1 stimulating the activation of SREBP1 (and to a lesser extent) the pentose phosphate pathway, but the precise mechanism remains undefined. HIF1 α is well known to upregulate glycolytic genes, but in preliminary experiments knockdown of HIF1 α in INS1 cells did not prevent the effects of glucose on expression of Pdk1 and other genes (Reviewer Fig.4).

How S6 kinase modulates PDH1e- α is currently unclear. It does not appear to affect Pdk1 expression (as chronic GAPDH inhibition with koningic acid activated mTORC1 and inhibited PDH, but did not upregulate Pdk1 expression). It also seems unlikely that S6 kinase has a direct physical association with PDK1 and/or PDH, or directly phosphorylates these enzymes because it would have to enter the mitochondrion to do so. Thus we think that this is probably an indirect effect, perhaps mediated by reversal of the changes in beta-cell metabolism. For example, it is possible that S6 kinase upregulates pyruvate dehydrogenase phosphatase (PDP1). We find that PDP1 mRNA levels are halved in diabetic islets/beta-cells but are normalised when euglycaemia is restored.

9. S6K inhibition at LG markedly augmented GSIS (Fig. 7d) suggesting that its effects on insulin secretion are mTORC1-independent. The experiments testing the effects of mTORC1 inhibition on glucose metabolism and insulin secretion are all based on a single pharmacological inhibitor of S6K. The authors should consider performing genetic manipulations of mTORC1 downstream targets, mainly S6K, to strengthen their interpretation of the findings.

The mTORC1 inhibitor PF-4708671 reduced pS6 in INS1 cells cultured at low glucose (Fig.7c), and augmented insulin secretion (Fig.7d). This argues that mTORC1 is partially activated even at low glucose in INS1 cells and its inhibitory effect on insulin release can be suppressed by inhibition of S6 kinase. This is consistent with the hypothesis that the effects of PF-4708671 are mTORC1-*dependent*.

We have now tested the effect of a second S6 kinase inhibitor, LY2584702 (Supplementary Fig. 6a,b). As for PF-4708671, we found that insulin content was unaffected, but insulin secretion in HG cells in response to 20mM glucose was enhanced. This data has now been added to the paper. Our data also suggest that there is not a linear relationship between insulin content and insulin secretion. It is well known that β -cells possess both readily releasable and stored pools of insulin granules, which may help account for this finding.

We now say: “A second inhibitor of S6K (LY2584702 [44]) produced a similar effect on glucose-stimulated insulin secretion in HG-cells (Supplementary Fig.6a,b)”.

10. Conditional KO of Tsc1/Tsc2 leading to constitutive activation of mTORC1 in beta-cells, hence mimicking the effects of diabetes, results in increased insulin content and secretion and improved glucose tolerance. This phenotype is preserved for a long period of time although late decline in beta-cell function is possible. How do the authors explain that activation of mTORC1 in the diabetic milieu rapidly impairs beta-cell function? These contradictory effects should be explained.

There is evidence that short-term activation of mTORC1 is beneficial for β -cells and insulin secretion but long-term activation is detrimental, as summarised in several reviews from the Maedler group (e.g. *Cell Metab* 27:314, 2018). For these reasons it is not unexpected that conditional KO of Tsc1/Tsc2 transiently enhances insulin secretion but later leads to beta cell failure (Bartolomé et al., 2014, Shigeyama et al., 2008, Rachdi et al., 2008). Additionally, activation of the mTORC1 pathway will alter all downstream targets (e.g. 4E-BP1, ULK-1), not just the S6 kinase pathway. Other studies suggest that overactivation of the S6 kinase pathway, specifically, may be detrimental to beta-cell function/metabolism (Elghazi et al., 2010, Yuan et al., 2017). It is also important to recognise that simply knocking down a gene (especially constitutively) is not always illuminating due to developmental effects and compensatory mechanisms. Furthermore, as we explain in our paper, activation of mTORC1 is not the only mechanism by which a glycolytic metabolite impairs insulin secretion. Different pathways appear to regulate insulin content and mitochondrial gene expression (TCA and ETC).

Minor comments

1. *Insulin secretion data shown in Supplemental Fig. 4c should be presented in a uniform manner as in the rest of the manuscript.*

We now present the insulin secretion data in Supplementary Fig. 4c in the same way as in the other figures. This now appears as Supplementary Fig. 3c.

2. *In the Discussion, the authors state that: "reducing GK activity may be therapeutic in T2D, evidence from people with a heterozygous inactivating mutation in GK provides support for this idea". Patients with MODY2 develop mild hyperglycemia and therefore rarely develop complications. This doesn't imply that moderate inhibition of glucokinase can ameliorate diabetes. I suggest toning down this statement.*

Our suggestion is to reduce glucokinase activity such that flux through the enzyme is comparable to that found at lower blood glucose levels. This would help prevent the deleterious effects on β -cell function produced by chronic hyperglycaemia and stabilise insulin content and secretion.

We argue that one of the reasons people with heterozygous GK mutations have mild glucose elevation is precisely because the mutation prevents the vicious spiral of hyperglycaemia leading to impaired insulin release, and thus to greater hyperglycaemia etc. Of course, if GK is completely or almost completely inhibited – as with homozygous GK mutations – then, as the reviewer points out, the patient does indeed have greatly increased glycaemia. The extent of GK inhibition is the crucial factor.

We have now rephrased the text to make this clearer and added reference to very recent papers (published after submission) that show genetic knockdown of GK in mice can prevent the loss of β -cell mass and insulin content in diabetes.

We now say: "Our results clearly demonstrate that excess glucose metabolism, rather than excess glucose, causes β -cell failure. Reducing glucose metabolism might therefore be an effective strategy to prevent the progressive decline in β -cell function that occurs in diabetes. The ability of mannoheptulose to prevent the effects of chronic hyperglycaemia suggests partial inhibition of glucokinase might be a viable strategy. Indeed, there is accumulating evidence that partial glucokinase inhibition both *in vitro* and *in vivo* can help preserve beta-cell function and mass in mouse models of diabetes^[62-64]. Although at first sight it may seem paradoxical to suggest reducing GK activity may be therapeutic in T2D, evidence from people with a heterozygous inactivating mutation in GK provides support for this idea. Despite mild fasting hyperglycaemia (~6.5mM), these patients require no medication, their hyperglycaemia does not progress and their prevalence of diabetic complications is not increased^[65,66]. Glucose homeostasis is simply maintained at a higher set point, resulting in mild asymptomatic hyperglycaemia. The counter-regulatory response to hypoglycaemia is also improved^[67]. This suggests that even as much as 50% loss of GK activity is not harmful. However, the aim of any therapy would be to restore GK activity in diabetes to that found at normal blood glucose levels in control beta-cells (and no further). Glucokinase is expressed in very few tissues (β -cells, liver, some neurones and endocrine cells). In all these cells, partial inhibition of glucokinase, leading to a reduction in G6P in the face of excess glucose, is likely to be beneficial."

Reviewer #2 (Remarks to the Author):

This is an impressive study that advances our understanding of the abnormal insulin secretion found in the diabetic state. Because of the thorough approach to the problem and the well-chosen measurement tools that were used, a variety of surprising findings emerged, which do not provide all of the answers but move us to a new level of understanding. Clarity of the thought processes and interpretations help the reader through complicated metabolic systems.

One of many strengths of the study was the use of two good beta cell model systems, INS-1 cells and diabetic betaV59M mice that have an inducible KATP channel mutation. The first important surprise was that the methyl ester form of pyruvate did not mimic the changes seen with high glucose concentrations. This helped focus attention on something important going on at the GAPDH level. The metabolomics data provided the key findings that there were large increases in F6P, F1, 6BP and F26BP but down-regulation of most TCA intermediates. This fit with the decrease in the activity of GAPDH and PDH, implicating the potential importance of some glycolytic metabolites downstream of glucokinase and upstream of GAPDH; that is, between G6P and pyruvate. It was particularly interesting that there was a marked decrease in the activity of GAPDH, despite a 3-fold increase in the GAPDH protein. This combination of findings led to the hypothesis that one or more of these metabolites caused upregulation of mTORC1 and downregulation of AMPK. Data were then generated that strengthened this hypothesis. The conclusions are well-supported by the data provided, and the discussion was well-written.

We thank the reviewer for their kind comments and are delighted that they find the study to be impressive, the interpretation clear and well-argued, and the paper clearly written.

Specific issues.

1. It would be helpful for readers to have a cartoon or some kind of diagram to show the key interactions that seem to be involved.

We have now included a cartoon as requested. It appears as Fig 10.

2. Not much attention was paid to beta cell death, which is OK because it was not studied. However, some mention might be helpful. Looking carefully at the beta cell mass data of Rahier and Henquin (Rahier J. Pancreatic alpha cell mass in European subjects with type 2 diabetes. Diabetologia. 2011;54(7):1720) It seems that the rate of beta cell death is very slow in spite of decades of hyperglycemia.

We agree that β -cell death is limited and cannot account for the reduction in insulin secretion found in diabetes. This is clearly shown in the Rahier paper, which we have now referenced. In many earlier papers, β -cell mass (i.e. the amount of β -cells) was quantified by insulin staining but as we and others have shown this is necessarily not a good indicator of β -cell mass because of the marked loss of insulin content in chronic hyperglycaemia (Brereton et al 2014, *Nature Commun* 5, 4639; Marselli, L. et al. 2014, *Diabetologia* 57, 362–365; Cinti et al, 2016, *The Journal of Clinical Endocrinology & Metabolism*, 101, 1044-1054). In addition, the rapid recovery of insulin secretion in subjects with type 2 diabetes following bariatric surgery, intensive insulin therapy or severe caloric restriction argues that loss of β -cells is not the cause their disease. This is discussed in the first paragraph of the Introduction.

3. The suggestion in the discussion that inhibition of glucokinase might be helpful when people have hyperglycemia does not seem to make sense. People with GK mutations do very well for two reasons. First, their glucose elevations are mild so they are largely spared from the glucose -induced complications. Second, they have good GSIS, which probably helps smooth out their control. With a GK inhibitor it seems glucose levels would climb and increase the glycemetic effects on complications.

Our suggestion is to reduce glucokinase activity such that flux through the enzyme is comparable to that found at lower blood glucose levels. This would help prevent the deleterious effects on β -cell function produced by chronic hyperglycaemia and stabilise insulin content and secretion. We envisage this would be combined with another therapy to help stabilise blood glucose levels while beta-cells recover.

We argue that one of the reasons people with heterozygous GK mutations have mild glucose elevation is precisely because the mutation prevents the vicious spiral of hyperglycaemia leading to impaired insulin release, and thus to greater hyperglycaemia etc. Of course, if GK is completely or almost completely inhibited – as with homozygous GK mutations – then, as the reviewer points out, the patient does indeed have greatly increased glycaemia. It is all about the dose (ie. the extent of GK inhibition).

We have now rephrased the text to make this clearer and added reference to very recent papers (published after submission) that show genetic knockdown of GK can prevent the loss of β -cell mass and insulin content in diabetes. Please see our reply to Reviewer 1, minor comment 2.

4. Throughout the paper, data are often described as n=3, sometimes as “in duplicate” and sometimes replicates are mentioned. Anyway, there is room for improvement. Please provide more information about how experiments with multiple replicates are handled.

We have now clarified this in the Methods section for each type of experiment.

We now say: “Unless otherwise stated, for islet experiments, n indicates the number of mice and for INS1 cell studies, n indicates the number of experiments. Most experiments had 2 or more (usually 3) technical replicates and the mean value of all replicates was taken as n=1. For oxygen consumption experiments, n indicates the number of replicates (wells). For Western blot experiments using islets, islets from multiple mice had to be pooled in order to obtain a sufficient amount of protein so we state both the number of experiments (n) and the total number of mice used.”

For clarity, we have removed reference to duplicates etc in the Figure legends.

5. Supp Fig. 1 Panels c: and d: Expressing secretion as ng/ml/islet/hour does not help us to understand what we really need to know, which is how much insulin is secreted per islet per hour.

This is a common way to express the data in the literature. However, as requested, we now express the data as ng/islet/hour. As the secretion experiments were performed in 1 ml of buffer this changes little.

6. *Supp Fig 2. There may be a better way to show these data. On my version, there are no asterix designating statistical significance. Also, there is no arrow pointing at F16BP.*

Thank you for pointing out that the arrow is missing. We forgot to remove the mention of an arrow from the figure legend when we removed it from the figure. We now present the data in Supplementary Fig.2 and 3 as Supplementary Data files as suggested below.

7. *Supp Fig 3. There must be some way to better highlight the important changes. I wonder if excel files would be helpful.*

As requested, we now include the data in Supplementary Figs 2 and 3 as Excel files. In addition, as all this data has been uploaded to a metabolomics database it can be easily accessed there.

8. *Supp Fig 4. It would be helpful to change the patterns and/or colors so that there is better discrimination between the vertical bars.*

We have now modified the Figure to give a clearer discrimination between the bars.

Reviewer #3 (Remarks to the Author):

Haythorne and colleagues examine the mechanisms by which chronic hyperglycemia and type II diabetes lead to reduced β -cell function and whether these pathological alterations are associated with mTORC1 activation. The authors claim that glycolytic metabolites downstream of hexokinase and upstream of GAPDH mediate the effects of diabetes and chronic hyperglycemia on β -cell metabolism. This happens through hyperactivation of mTORC1, which leads to alterations in insulin secretion and oxidative phosphorylation. In addition, the authors show that hyperglycemia inhibits GAPDH and pyruvate dehydrogenase (PDH) activities. The authors use control or Db islets, and INS1 cells cultured in low (LG) or high glucose (HG) to recapitulate diabetes and chronic hyperglycemia. Overall, the authors support the idea that gradual damage of β -cell metabolism, induced by increasing hyperglycemia, promotes diabetes development and suggest that reducing glycolysis at the level of hexokinase may slow this progression.

General comments:

1. *While the concept proposed by the authors is interesting, the mechanisms by which glycolysis controls chronic hyperglycemia remain unidentified. The study is preliminary but provides a hint on how potential glycolytic intermediates upstream of GAPDH and downstream of hexokinase support chronic hyperglycemia and diabetic state. The take-home message of the paper is confusing and needs to be further elucidated. The authors should focus on clearly identifying the identity of the metabolite or metabolic enzyme involved in controlling insulin secretion and insulin content in INS1 cells or diabetic islets.*

We disagree that the take-home message is confusing. Indeed, Reviewer 2 specifically comments on the clarity of interpretation and discussion. However, we have now included a cartoon to help the reader (Fig.10).

To further pinpoint the identity of the metabolite involved in regulating the effects of chronic hyperglycaemia, we have performed additional experiments. First, we used 6-AN to exclude a role for the pentose phosphate pathway. Second, we carried out knockdown experiments of glycolytic enzymes upstream of GAPDH to further pinpoint the metabolite(s) involved. These data are now added to the paper.

We now say: 'We excluded a role for the pentose phosphate pathway as inhibition of 6-phosphoglucanate dehydrogenase with 6-AN with did not prevent the effect of hyperglycaemia on insulin secretion, insulin content or gene expression (Supplementary Fig.2)'.

Please also see reply to Reviewer 1, point 1.

2. The data with the S6K1 inhibitor (Fig. 7d,e) are not convincing. Treatment with S6K1 inhibitor slightly increases insulin content in HG cells. This suggests that the contribution of S6K in the control of hyperglycemia is relatively negligible. The authors should use rapamycin (mTORC1 inhibitor) or even Akt inhibitor since Akt has also been proposed to respond to glucose metabolism.

Rapamycin is not selective and inhibits both mTORC1 and mTORC2 when used chronically (Sarbassov et al. *Mol Cell*. 22:159-68, 2006). We have therefore used another specific inhibitor of S6 kinase (LY2584702). Treatment with this inhibitor enhanced insulin secretion in HG cells but did not prevent the loss of insulin content. These data now are included as Supplementary Fig 6b. PF-4708671 partially prevented the reduction in insulin content in HG cells but was unable to reverse the loss of insulin content in diabetic islets. These data argue that the effects of chronic hyperglycaemia on insulin content and β -cell metabolism are regulated by different mechanisms. We have revised the text to make it clearer that we think the mTORC1/S6 kinase pathway regulates glycolytic gene expression, but that it does not appear to regulate insulin gene expression or expression of many genes/proteins involved in mitochondrial metabolism. As discussed in the reply to Reviewer 1, the regulation of insulin content by hyperglycaemia is the subject of a separate paper.

We now say: A second inhibitor of S6K (LY2584702 [44]) produced a similar effect on glucose-stimulated insulin secretion in HG-cells (Supplementary Fig.6a,b).

And we also say: "Taken together, the indicate that S6K signalling is involved in regulation of glycolytic but not mitochondrial (other than Pdk1) gene expression in response to chronic hyperglycaemia.

3. A previous study has linked dihydroxyacetone phosphate (DHAP) to mTORC1 activation. So, does the supplementation of dihydroxyacetone decreases insulin content and secretion in LG cells? Of course, to get this rescue to work, the authors should express triose kinase (TKFC) in INS1 cells to make sure that exogenous dihydroxyacetone can get converted into DHAP to signal mTORC1 signaling.

The ideal experiment would be to permeabilise the plasma membrane to enable DHAP to enter the cell but, were we to do so, we would then simultaneously lose other metabolites such as NAD⁺, NADH and ATP. While these could be included in the medium with DHAP, we do not know the correct level at which to supply them. It should also be remembered that DHAP can be rapidly converted to GA3P and F16BP, so this experiment could not distinguish between them.

In addition, as it is not known how DHAP activates mTORC1, this experiment is really part of a much larger set of experiments that lie outside the scope of the present paper and will be addressed in a subsequent one.

4. Based on the metabolomics data presented in this manuscript, it is hard to pinpoint the specific metabolite responsible for the diabetic phenotype. DHAP and glyceraldehyde 3-phosphate (GA3P) have the exact same mass. Therefore, it is surprising that the authors could differentiate them by classic LC-MS. For example, I am not convinced that GA3P reads presented in Fig. 4a are not a mixture of DHAP/GA-3P. How did the authors ensure that GA3P is not DHAP+GA3P? Moreover, if koningic acid (KA) inhibits GAPDH activity, should not we observe an accumulation of GA3P levels upon LG+KA (20 mM Glucose) condition?

We thank the reviewer for pointing this out. Indeed, it is correct that the GA3P measurements reported in Fig 4a are for a mixture of DHAP/GA3P. It is possible for some highly polar and ionic structural isomers to be resolved chromatographically using this method as shown for other sugar monophosphates. However, in the case of DHAP/GA3P we performed follow up experiments using authentic standards and determined that DHAP/GA-3P co-elute and are not resolved. Therefore, the abundance of GA3P reported in Figure 4a is for the mixture of both metabolites. The figures have been re-drawn to reflect this and the text has been amended. Please note that we also measured DHAP biochemically, in a separate assay (Figure 4a).

5. In addition to koningic acid, other inhibitors of glucose metabolism could be used to methodically study the contribution of key glycolytic branches in the control of hyperglycemia and diabetes. Since intermediates of the pentose phosphate pathway are increased upon high glucose condition, targeting G6PD (6-aminonicotinamide) to decrease 6-phosphogluconate and transketolase (oxythiamine) to decrease sedoheptulose 7-phosphate could be employed to assess the effects of these inhibitors on the insulin secretion and content in LG cells.

We have tested the effect of inhibiting G6PD with 6-AN, as suggested. We found that 100µM 6-AN had little or no effect on the changes in glycolytic gene expression, or insulin secretion, induced by chronic hyperglycaemia, This helps exclude a mechanistic role for pentose phosphate pathway intermediates.

We now say: “We excluded a role for the pentose phosphate pathway as inhibition of 6-phosphogluconate dehydrogenase with 6-AN with did not prevent the effect of hyperglycaemia on insulin secretion, insulin content or gene expression (Supplementary Fig.2).”

We also now include a number of knockdown experiments which further localise the key metabolite(s) mediating the effects of hyperglycaemia. Please see reply to Reviewer 1, point 1 for details.

Specific comments:

1. The authors should reduce the number of main Figures (The manuscript contains now nine main figures, and the figure could be condensed and presented more succinctly. For example, figure 2, which shows negative data, could be moved to the supplemental item.

We believe that Figure 2 is important and prefer to retain it. It is not negative data in the usual sense as it conveys a key message. Indeed Reviewer 2 specifically comments on its importance –‘the first important surprise was that the methyl ester form of pyruvate did not mimic the changes seen with high glucose concentrations’. As Reviewer 2 also says, ‘This helped focus attention on something important going on at the GAPDH level’.

2. AMPK phosphorylation does not necessarily reflect AMPK activity. Therefore, in addition to p-AMPK, the authors should present phosphorylation of AMPK substrates (p-ACC (S79)/ACC or p-RAPTOR (S792)/RAPTOR).

We now include data for p-Raptor (Figure 5a,c,f,h). The data look similar to that for p=AMPK.

3. Fig. 5d: P-S6/S6 signal should be improved.

A different blot is now shown.

4. Fig. 6c: The p-S6 signal does not appear to be increased upon 20 mM glucose compared to 2 mM glucose. The authors should provide a better representative western blot.

We agree that the difference in the Western blot shown does not look very strong, but there is a clear difference in the densitometry signal (Fig. 6e).

There was always an increase in pS6 in HG cells and diabetic islets (compared to controls) at 2mM glucose. In control islets there was also always an increase in pS6 when glucose was acutely elevated from 2 to 20mM. The response to acute glucose elevation was variable in LG cells – sometimes there was a clear increase and sometimes the increase was relatively small. It is possible this is because the INS1 cells are proliferating. However, this does not affect our studies, which focus on the effects of chronic glucose.

5. Other key markers of mTORC1 signaling should be presented, such as p70S6 kinase and 4E-BP1 phosphorylation.

We now present data on 4E-BP1 phosphorylation (Figure 5a,e,f,j). The data look similar to that for pS6.

REVIEWERS' COMMENTS

Reviewer #1 (Remarks to the Author):

The authors adequately addressed my concerns.

Reviewer #2 (Remarks to the Author):

The first version of this paper was very strong because the Oxford team used a systematic approach to demonstrate a variety of metabolic changes in beta cells exposed to hyperglycemia that were accompanied by impaired insulin secretion. It was fascinating that they could make a case for the importance of some glycolytic intermediates downstream of glucokinase and upstream of GAPDH. These changes were accompanied by increased activation of mTORC1 and inhibition of AMPK. There may also be something important going on with the two aldolase enzymes. It seems that these findings are leading to improving our understanding of what is causing the marked abnormalities of mitochondrial function that impair insulin secretion, one example being inhibition of PDH (pyruvate dehydrogenase). These abnormalities may well turn out to be fundamental to our understanding of the pathogenesis and treatment of diabetes, not only T2D but also T1D.

The revision is stronger, in part because of the excellent suggestions by reviewer #1, to carry our knockdowns experiments on some key components of the puzzle.

All of my concerns have been addressed. I would like to compliment those who performed this excellent study, which takes these important questions to the next level.

Reviewer #3 (Remarks to the Author):

The authors have addressed most of my concerns.

I would like to congratulate the authors on this revised manuscript.

NCOMMS-21-48720A: REPLY to the REVIEWERS

We thank all the reviewers for their very positive comments on our study. No further actions were requested by the reviewers.

*Reviewer #1 (Remarks to the Author):
The authors adequately addressed my concerns.*

No action is required

*Reviewer #2 (Remarks to the Author):
The first version of this paper was very strong because the Oxford team used a systematic approach to demonstrate a variety of metabolic changes in beta cells exposed to hyperglycemia that were accompanied by impaired insulin secretion. It was fascinating that they could make a case for the importance of some glycolytic intermediates downstream of glucokinase and upstream of GAPDH. These changes were accompanied by increased activation of mTORC1 and inhibition of AMPK. There may also be something important going on with the two aldolase enzymes. It seems that these findings are leading to improving our understanding of what is causing the marked abnormalities of mitochondrial function that impair insulin secretion, one example being inhibition of PDH (pyruvate dehydrogenase). These abnormalities may well turn out to be fundamental to our understanding of the pathogenesis and treatment of diabetes, not only T2D but also T1D.*

The revision is stronger, in part because of the excellent suggestions by reviewer #1, to carry our knockdowns experiments on some key components of the puzzle.

All of my concerns have been addressed. I would like to compliment those who performed this excellent study, which takes these important questions to the next level.

No action is required. We thank the reviewer for their kind comments on our study.

*Reviewer #3 (Remarks to the Author):
The authors have addressed most of my concerns.
I would like to congratulate the authors on this revised manuscript.*

No action is required. We thank the reviewer for their kind comments on our study.